# Q4ddPCR: a flexible, 4-target assay for high-resolution HIV reservoir profiling

Rachel Scheck [1,2], Mark Melzer [3], Gregory Gladkov [2], Louise Leyre[2], Adam R. Ward [2], Daniel B. Reeves [4], Naomi Perkins[1], T. Thinh Huynh [2], Deborah K. McMahon[5,6], Ronald J. Bosch[7], Bernard J. Macatangay [5,6], Joshua C. Cyktor [5], Joseph J. Eron [8], John W. Mellors[5], Rajesh T. Gandhi[9], Lisa Buchauer [3], R. Brad Jones [2,10] ✉ & Christian Gaebler [1,10] ✉

Precise and scalable quantification of the intact HIV reservoir is critical for advancing curative strategies. Current reservoir assays, such as the intact proviral DNA assay (IPDA), are limited by quantification failures or misclassification of defective proviruses due to HIV sequence heterogeneity. Q4ddPCR is a modular, droplet digital PCR simultaneously targeting four conserved regions in the HIV genome to improve specificity, reduce quantification gaps, and provide multi-layered readouts. It comprises two configurations: one fully based on Q4PCR primer/probes and one combining IPDA with *gag* and *pol* primer/probes from Q4PCR. We benchmark Q4ddPCR against 3650 near full-length proviral sequences from 13 virally suppressed people with HIV (PWH) generated by Q4PCR. Q4ddPCR closely matches sequence-confirmed reservoir measurements, and multi-probe readouts reveal clonal reservoir dynamics not detectable by IPDA. Q4ddPCR enables intact reservoir quantification in 95% of samples across four independent cohorts and in 16 PWH, strongly correlates with viral outgrowth. In longitudinal samples from 42 participants over the first 4.5 years on antiretroviral therapy (ART), Q4ddPCR reports lower proviral frequencies and a steeper decline in intact proviral DNA compared to IPDA. Collectively, our findings confirm key predictions from mathematical modeling, demonstrating that multi-target assays improve specificity and more accurately capture intact reservoir dynamics.

The persistence of genetically intact HIV proviruses in long-lived, clonally expanded CD4+ T cells remains the principal barrier to a broadly applicable HIV cure[1]. Although the majority of integrated proviruses are genetically defective and incapable of reactivation, a small subset of genetically intact proviruses can reinitiate viral replication if antiretroviral therapy (ART) is interrupted[1,2]. Accurate quantification of this intact reservoir is essential for understanding the mechanisms of HIV persistence and evaluating curative strategies[1].

[1]Laboratory of Translational Immunology of Viral Infections, Department of Infectious Diseases and Critical Care Medicine, Charité-Universitätsmedizin Berlin, Berlin Institute of Health, Berlin, Germany. [2]Division of Infectious Diseases, Department of Medicine, Weill Cornell Medicine, New York, NY, USA. [3]Laboratory of Systems Biology of Infectious Diseases, Charité-Universitätsmedizin, Berlin, Germany. [4]Vaccine and Infectious Disease Division, Fred Hutchinson Cancer Center, Seattle, WA, USA. [5]Division of Infectious Diseases, University of Pittsburgh, Pittsburgh, PA, USA. [6]Department of Infectious Diseases and Microbiology, University of Pittsburgh School of Public Health, Pittsburgh, PA, USA. [7]Center for Biostatistics in AIDS Research, Harvard TH Chan School of Public Health, Boston, MA, USA. [8]Division of Infectious Diseases, University of North Carolina, Chapel Hill, NC, USA. [9]Infectious Diseases Division, Massachusetts General Hospital, Harvard Medical School, Boston, MA, USA. [10]These authors contributed equally: R. Brad Jones, Christian Gaebler. ✉e-mail: rbjones@med.cornell.edu; christian.gaebler@charite.de

A variety of assays have been developed to measure the HIV reservoir, each with distinct advantages and limitations. Quantitative viral outgrowth assays (QVOA) detect inducible, replication-competent virus, providing a functional measure and a minimal estimate of the reservoir size. However, QVOA is labor-intensive, costly, requires large cell inputs, and underestimates reservoir size due to limited viral inducibility with a single round of stimulation[3].

PCR-based approaches, by contrast, offer greater scalability and require fewer cells. These assays target conserved regions of the HIV genome to quantify proviral DNA[4]. However, single-target PCR methods substantially overestimate HIV reservoir size by including defective proviruses, as they cannot distinguish between intact sequences and defective sequences with deletions or mutations[4].

To address this, the Intact Proviral DNA Assay (IPDA) was developed as a high-throughput droplet digital PCR (ddPCR) method to detect intact proviruses through amplification of two conserved regions of the HIV genome: the packaging signal ($\Psi$) and the Rev-responsive element (RRE) in *env*, which were strategically selected based on the degree of sequence conservation in near full-length proviral genome alignments. To further discriminate and exclude hypermutated proviruses, a non-fluorescent ("dark") probe is included[5]. For IPDA, individual proviruses are encapsulated within droplets and classified as 'intact' if both targets are co-detected. In contrast, a provirus is classified as 'defective' if only one target is detected (with a correction applied for DNA shearing)[5]. While IPDA effectively excludes approximately 93-97% of defective proviruses, the accuracy is limited by misclassification of proviruses with internal deletions or mutations outside of the targeted regions. Only 51-70% of the detected proviruses using simultaneous detection of an *env*- and $\Psi$-target are truly intact by near full-length genome sequencing[6,7]. This is significant given the differential decay rates of intact and defective proviruses: the intact reservoir has a half-life of approximately one to four years over the first seven years of ART (and subsequently plateaus or even increases[8–10]), whereas the defective reservoir remains relatively stable over time[7,11–14]. As a result, the misclassification of defective genomes as "intact" proviruses can lead to an underestimation of reservoir decay and obscure the effects of clinical interventions aimed at reducing the HIV reservoir. Mathematical modeling highlighted that 3- or 4-target approaches in PCR-based reservoir-quantification methods may result in higher specificity of intact reservoir quantification[7].

While early implementations of such multiprobe approaches were limited to quantitative PCR (qPCR) platforms or shared fluorophores with different signal intensities, advances in digital PCR (dPCR) technology now enable true (droplet) dPCR-multicolor detection, such as the Rainbow assay[15–19].

Another substantial challenge of the IPDA and related 2-target approaches is the reported assay failure rate of approximately 12-30% in HIV subtype B, largely due to intra- and interindividual HIV sequence heterogeneity leading to primer/probe mismatches[6,13,17,20–24]. Strategies to overcome these failures have included the use of secondary backup primer/probe sets or individualized primer/probe designs based on additional proviral sequencing[17,20,23]. While individualized primer/probes can rescue detection in some cases, they introduce significant challenges related to scalability, cost, labor, and the need for additional sample material, limiting their broader applicability in clinical research settings. Q4PCR, a 4-target qPCR method combined with near full-length sequencing, addresses some of these limitations by increasing the number of genomic targets used for classification[15,6]. It improves specificity and allows confirmation of sequence integrity and clonality on a molecular level. However, scalability is constrained by the assay's labor-intensive workflow, which involves limiting-dilution long-distance PCR, 4-target qPCR, and near full-length sequencing[15,6]. In particular, the long-distance PCR (approximately 9 kb) is inherently inefficient, resulting in reduced sensitivity for proviral detection and

ultimately contributing to an underestimation of the intact reservoir[7,25,26].

With the aim of developing a new standard for high-throughput, cost-effective, sensitive, and precise reservoir quantification, we developed a novel droplet digital PCR assay called Q4ddPCR that draws on the strengths of both IPDA and Q4PCR. By incorporating four conserved targets in the HIV genome, Q4ddPCR is designed to enhance both sensitivity and specificity. We also leverage the ability to consider multiple different combinations of target detection to mitigate the impact of detection failures resulting from interindividual sequence heterogeneity. We first validated the assay using longitudinal samples from 13 virally suppressed people with HIV (PWH) for whom near full-length proviral sequences had been previously obtained using Q4PCR[6,15,27,28]. This allowed for direct benchmarking and a standardized analysis framework for reporting intact proviral genomes. To further evaluate clinical utility, we applied Q4ddPCR to three distinct cohorts. In the KOHIVI cohort, we performed a direct, side-by-side comparison of Q4ddPCR and the IPDA on matched samples to assess differences in quantification accuracy and assay robustness. We also examined the relationship between molecular and functional reservoir measurements in a subset of 16 ART-suppressed PWH, comparing Q4ddPCR and IPDA with the quantitative viral outgrowth assay (QVOA). Finally, we implemented Q4ddPCR in a longitudinal study of 42 PWH enrolled in ACTG trials who later entered a prospective cohort study (ACTG A5321) to characterize intact reservoir dynamics over the first 4.5 years of ART[29]. This included assessing assay performance over time and testing the hypothesis - supported by prior mathematical modeling - that multi-target approaches provide a more accurate estimate of intact reservoir decay compared to 2-target assays.

## Results

### Design and optimization of the Q4ddPCR assay

We developed a 4-target droplet digital PCR assay (Q4ddPCR) for high-throughput quantification of the intact HIV reservoir, incorporating primer/probe sets targeting conserved regions in *env*, $\Psi$, *gag*, and *pol*. Two assay versions were evaluated: one with primer/probes entirely based on Q4PCR primer/probe sequences (Q4-based Q4ddPCR), and another combining IPDA-derived primer/probes for $\Psi$ and *env* with Q4PCR-derived primer/probes for *gag* and *pol* (IPDA-based Q4ddPCR, Fig. 1a, Supplementary Table 1). The two versions differ in the sequence of the $\Psi$-targeting primer/probes and fluorophore configuration for $\Psi$ and *gag* probes (Supplementary Fig. 1)[5,15]. We included a separate reaction to quantify cell equivalents and to correct for shearing using the RPP30-reaction that is used for IPDA[5]. We optimized primer/probe concentrations and cycling conditions using DNA from J-Lat 6.3 cells harboring a single integrated intact provirus per cell (Fig. 1b, c, Supplementary Fig. 1c). We systematically evaluated annealing temperatures and number of PCR cycles, identifying 55 °C and 60 cycles as the conditions that yielded the cleanest cluster separation with minimal intermediate fluorescence values.

We next assessed linearity and dynamic range following ddPCR guidelines[30]. Three independent 10-step dilution series were prepared using DNA from spiked-in J-Lat 6.3 cells into PBMCs from people without HIV (PWoH), spanning 0.625 to 3000 copies, and run in eight replicates per dilution. Both Q4-based and IPDA-based Q4ddPCR showed good linearity, with $R^2 > 0.99$ and a median slope of 1.0 (range: 0.88–1.18) across all intact-defining readouts and single-target measurements (Supplementary Fig. 2). Coefficients of variation (CV%) were comparable between dilution curves and increased at lower input copy numbers as expected (Supplementary Table 2). To assess assay variability in samples from PWH with low reservoir sizes, we repeated the full workflow from CD4+ T-cell isolation to DNA extraction and Q4ddPCR 7 to 12 times in four participants with geometric mean reservoir sizes of 5–43 intact proviruses per $10^6$ CD4+ T cells. Q4-based and IPDA-based Q4ddPCR were highly concordant ($\rho = 0.75$; 95% CI,

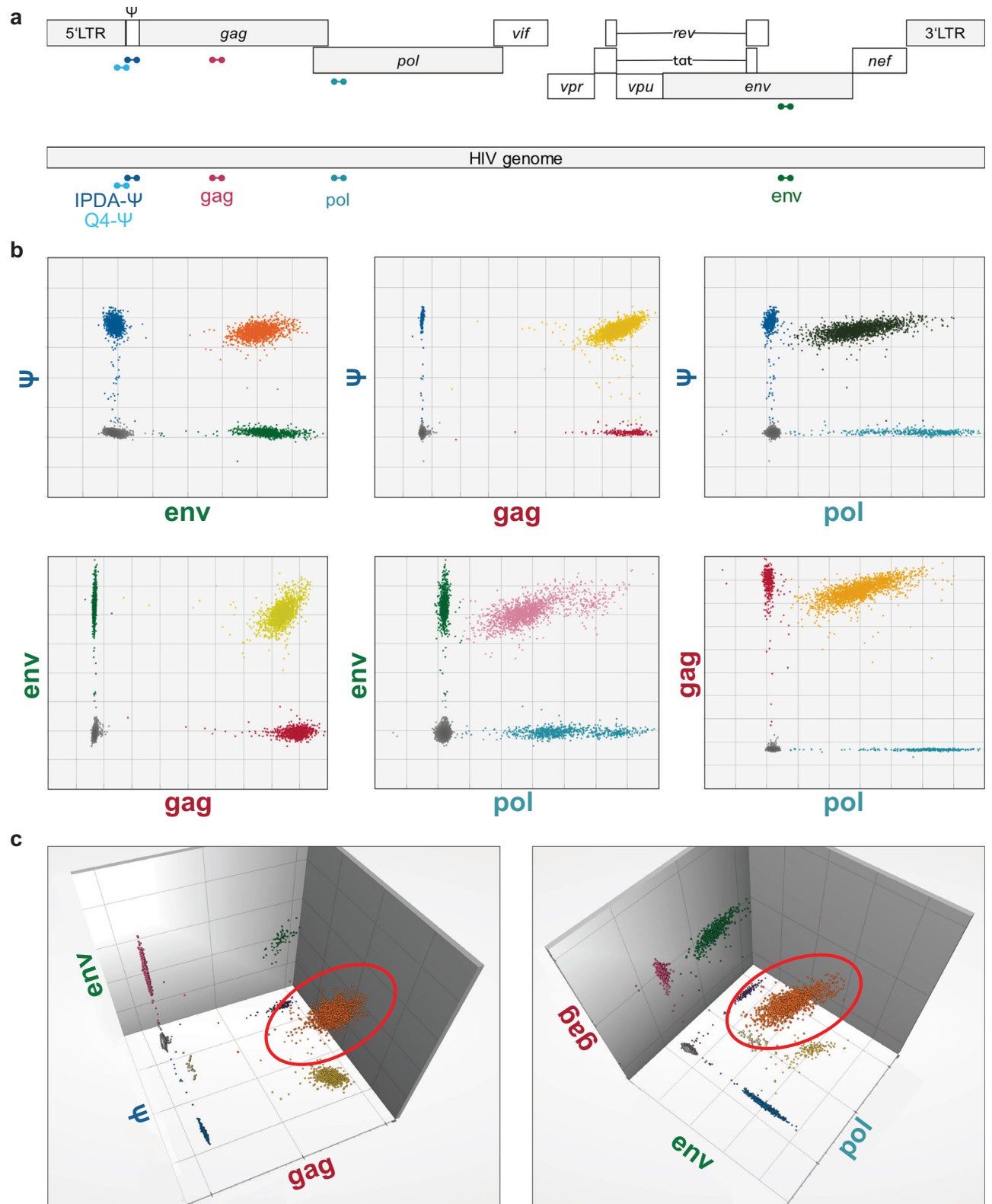

**Fig. 1 | Development and optimization of the Q4ddPCR assay. a** Schematic representation of Q4ddPCR target regions within the HIV genome: *env* (green), packaging signal (*Ψ*; light and dark blue), *gag* (red), and *pol* (turquoise). Q4ddPCR utilizes two distinct primer/probe sets which are referred to as Q4-based Q4ddPCR (all primers/probes derived from Q4PCR) and IPDA-based Q4ddPCR (primers/probes for *env* and *Ψ* derived from IPDA; *gag* and *pol* derived from Q4PCR). The sets differ in the sequences of the *Ψ*-targeting primers and probes, as well as the fluorophores attached to *Ψ* and *gag* probes. Probe binding sites are indicated. Representative two-dimensional (**b**) and three-dimensional (**c**) IPDA-based Q4ddPCR plots from genomic DNA of J-Lat 6.3 cells. Double-positive droplets are color-coded as follows: orange (*env-Ψ*), yellow (*Ψ-gag*), black (*Ψ-pol*), light green (*env-gag*), pink (*env-pol*), and light orange (*gag-pol*, b). Triple-positive droplets are highlighted with red circles (**c**).

0.07–0.95), with 84% of replicate measurements for individual targets or intact-defining readouts falling within a 2-fold range of the mean (Supplementary Fig. 3a).

The limit of detection (LoD), defined as the lowest dilution at which ≥95% of replicates were positive, was 10 copies for Q4-based Q4ddPCR and 5 copies for IPDA-based Q4ddPCR (Supplementary Fig. 3b). In 94 (Q4-based) and 157 (IPDA-based) negative-control reactions using PBMC DNA from PWoH, no false triple- or quadruple-positive droplets were detected. Only a single IPDA-based Q4ddPCR reaction contained one false *env-gag*-positive droplet. Accordingly, the limit of blank (LOB) for intact readouts (4-color, *env-gag-Ψ*, *env-Ψ-pol* and *env-Ψ*) was 0 copies for both assay versions, except for *env-gag* in the IPDA-based assay (LOB = 0.008 copies). For single-target readouts, LOB values were as follows: Q4-based Q4ddPCR: *env* 0.01, *Ψ* 0.49, *gag* 0.13, *pol* 0.27 copies; IPDA-based Q4ddPCR: *env* 0.32, *Ψ* 0.44, *gag* 0.06, *pol* 0.77 copies.

Next, we evaluated the inclusion of a fifth probe targeting the 5′ LTR region to quantify total HIV DNA[17,31,32]. However, substantial fluorescence spillover was observed in both standard material and participant samples (Supplementary Fig. 4a, b). The magnitude of this spillover overlapped with the expected signal range of true *pol*-positive droplets, making computational compensation impractical, as it would artificially reduce *pol* detection. Finally, we compared total HIV DNA quantification by Q4ddPCR, calculated by summing single-target positive proviruses with correction for overlap, with total HIV DNA measured by a single-probe 5′LTR ddPCR assay in 28 PWH (Supplementary Table 3). Q4ddPCR total HIV DNA strongly correlated and was highly concordant with 5′LTR measurements (Q4-based: Spearman $r = 0.87$, $p < 0.0001$, $\rho_c = 0.83$, 95% CI 0.67 − 0.92; IPDA-based: $r = 0.86$, $p < 0.0001$, $\rho_c = 0.80$, 95% CI 0.61 − 0.90) (Supplementary Fig. 4c).

Thus, the 4-probe Q4ddPCR assay achieves robust total HIV DNA quantification and avoids fluorescence spillover-related artifacts that can impair assay specificity.

## Performance evaluation of Q4ddPCR using previously characterized HIV reservoir samples

To rigorously validate the Q4ddPCR assay, we leveraged longitudinal samples from a cohort of 13 virally suppressed individuals that had previously been characterized using Q4PCR (Fig. 2a, Supplementary Table 4)[6,15]. These participants had been enrolled in studies evaluating broadly neutralizing antibodies (bNAbs), and while all were receiving ART at the initial sampling time point, ART was interrupted at the second time point as part of an analytical treatment interruption (ATI) protocol. Despite ART interruption, plasma viral loads remained below the limit of detection at both time points[27,28]. In our earlier Q4PCR analysis, all samples positive for at least two targets underwent sequencing, yielding a total of 3,650 proviral sequences including 558 intact proviral genomes[6,15]. Utilizing these deeply characterized samples enabled precise benchmarking of Q4ddPCR-derived results against high-resolution, sequence-confirmed reference data.

To assess assay performance, we compared different Q4ddPCR readouts - categorized by proviruses positive for 1-, 2-, 3-, or all 4-targets - to the frequency of intact proviruses per $10^6$ CD4$^+$ T cells previously measured by Q4PCR (Fig. 2a, b; Supplementary Fig. 5a). Using the 4-target readout, proviral frequencies measured by Q4ddPCR showed strong correlation with Q4PCR-derived intact proviral frequencies (Spearman's $r = 0.72$, $p < 0.0001$ for Q4-based Q4ddPCR; Spearman's $r = 0.62$, $p = 0.001$ for IPDA-based Q4ddPCR) albeit with slightly higher median values (2-fold for Q4-based; 3-fold for IPDA-based Q4ddPCR, Fig. 2b, c, Supplementary Fig. 5a, c). In contrast, lower-order readouts (1-, 2-, or 3-target combinations) systematically overestimated reservoir sizes relative to Q4PCR. Specifically, median proviral counts were 40- and 43-fold higher using 1-target readout, 15- and 18-fold higher using 2-targets, and 3- and 5-fold higher using 3-target combinations for Q4- and IPDA-based

Q4ddPCR assays, respectively (Fig. 2b, Supplementary Fig. 5a). Notably, among the 2-target readouts, the *env-Ψ* combination measured using IPDA-based Q4ddPCR showed the strongest correlation with Q4PCR-derived intact proviral frequencies (Supplementary Fig. 5c).

A known limitation of PCR-based methods for quantifying intact proviruses, including IPDA, is amplification failures caused by HIV sequence variability, giving rise to mismatches in primer/probe binding regions[20]. To address this, we tested both Q4- and IPDA-based primer/probe sets on the same samples. Using both primer/probe sets successfully restored *Ψ* signal in 3 of 4 participants with amplification failures: Two IPDA-*Ψ* failures were restored using the Q4-*Ψ*, one Q4-*Ψ* failure was restored using IPDA-*Ψ* (Fig. 2d, Supplementary Fig 4d).

Importantly, since Q4PCR includes sequencing of proviruses amplified with the same target combinations as Q4ddPCR, we were able to track individual sequence-confirmed clones over time. For example, in participant 9242, the dominant intact clones were *env-Ψ*-positive. However, sequence-confirmed intact *env-Ψ*-negative clones which declined over time had also been previously identified by Q4PCR. These clones were captured by alternative Q4ddPCR readouts such as *env-gag* (Fig. 2d, Supplementary Fig. 5b, d) and would have been overlooked by standard IPDA, underscoring how the flexibility to use multiple primer/probe combinations in Q4ddPCR can effectively overcome assay limitations imposed by HIV sequence variability.

Collectively, these results demonstrate that Q4ddPCR closely approximates intact proviral frequencies measured by Q4PCR. By leveraging multiple conserved targets and alternative primer/probe sets without requiring a sequencing step, Q4ddPCR combines high-throughput capability with the flexibility needed to overcome sequence heterogeneity within the reservoir. Furthermore, in some cases, Q4ddPCR enables tracking of individual intact clones that would otherwise be missed by exclusively looking at *env-Ψ*-positive proviruses.

## Standardized decision tree for reporting intact proviruses by Q4ddPCR

To assess how accurately different Q4ddPCR readouts detect intact proviral DNA, we compared Q4ddPCR results to matched proviral sequences from all 13 participants obtained by Q4PCR. These sequences were derived from amplicons that have been tested positive in the Q4PCR qPCR step using the same primer/probe combination as Q4ddPCR[6,15]. This allowed us to benchmark Q4ddPCR detection against a sequence-confirmed reference framework. For each Q4ddPCR probe combination, we calculated sensitivity as the proportion of sequence-confirmed intact proviruses detected and specificity as the proportion of defective sequences correctly excluded. This approach reflects both the likelihood of detecting truly intact proviruses and the accuracy with which intactness is assigned by each target combination.

While detection frequencies expectedly decline with additional targets, the specificity of accurately identified intact proviral genomes increases from 42% for 1-target detection to 95% for 4-target (4-colors) readout (Fig. 3a, Supplementary Table 5).

However, we observed substantial inter-individual variability in the diagnostic performance of different target combinations (Fig. 3b, Supplementary Fig. 6). In participant 5111, 4-color positive proviruses could not be detected, and *env-gag-Ψ* and *env-Ψ* positives showed equal proviral counts with identical sensitivity and specificity. In participants with a more clonal reservoir (e.g., 5104), the 4-target readout maintained high sensitivity while offering substantially higher specificity compared to 2-target readouts (57% vs. 38%). In participants 9244 and 9255 where sequence mismatches in the Q4-based *Ψ* primers and probes result in amplification failures; alternative readouts (*env-gag* or *env-gag-pol*) resulted in sensitive and specific estimates for intact proviral frequencies (Fig. 3b, Supplementary Fig. 6).

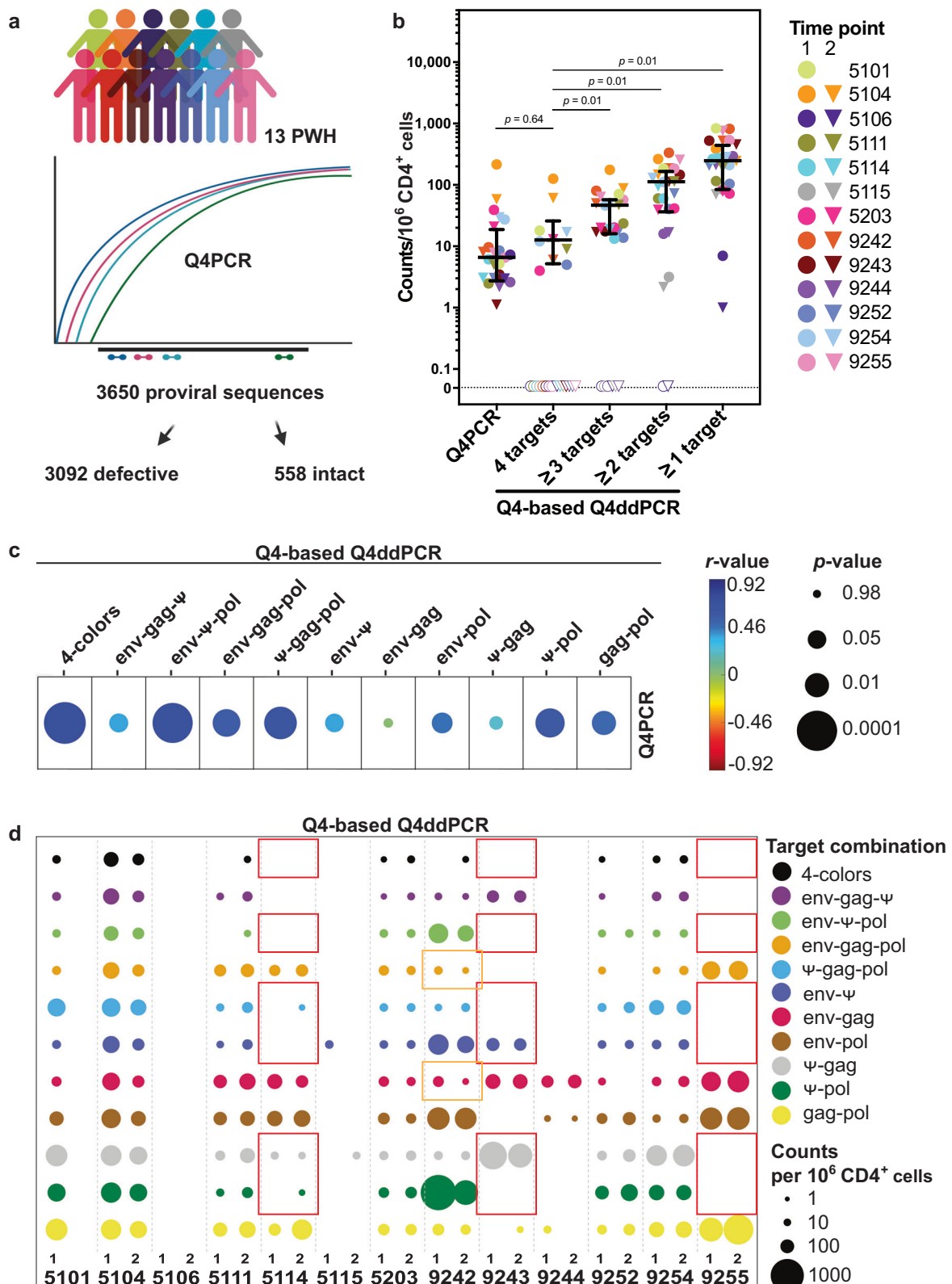

Overall, these findings demonstrate that no single target combination performs optimally across all individuals, underscoring the need for a structured analysis framework to fully leverage the flexibility of our 4-target assay.

To address this, we developed a decision tree for classifying intact proviruses based on the specificity-focused performance of different Q4ddPCR readouts (Fig. 3c). Because accurate reservoir decay estimation depends on high specificity[7], we prioritized reporting 4-target positive proviruses when detected, followed by *env-Ψ*-including 3-target combinations. The *env*-target showed the highest sensitivity and negative predictive value in the Q4PCR-derived sequence data (Supplementary Table 5). Consequently, target combinations lacking *env* were excluded from the definition of intactness.

**Fig. 2 | Validation of Q4ddPCR on longitudinal samples from 13 people with HIV (PWH). a** Q4ddPCR was applied to longitudinal samples from 13 PWH previously characterized using Q4PCR. Q4PCR combines a 4-target qPCR with near full-length genome sequencing and shares the primer/probe sequences with Q4-based Q4ddPCR. All samples positive for ≥2 targets by Q4PCR had previously been sequenced, resulting in 3650 proviral sequences for these 13 PWH. Created in BioRender. Scheck, R. (2026) https://BioRender.com/h0nw368. **b** Proviral counts per $10^6$ CD4$^+$ T cells measured by Q4ddPCR across different readout combinations, compared to intact proviral counts from Q4PCR. Average counts from Q4-based Q4ddPCR are grouped by the number of detected targets (1–4). Each symbol represents an individual sample ($n = 25$); circles indicate the first time point ($n = 13$), triangles the second ($n = 12$). Samples from the same participant are color-matched. Samples with zero detected proviruses for a given target number are plotted with transparent symbols on the dotted line. Medians and interquartile ranges are shown. Statistical comparisons used the two-sided Wilcoxon signed-rank test with post hoc correction for multiple comparisons. **c** Two-sided Spearman correlation between Q4-based Q4ddPCR readouts and reservoir size as previously measured by Q4PCR. Correlations are shown for different Q4ddPCR target combinations (columns) with intact proviruses per $10^6$ CD4$^+$ T cells measured by Q4PCR. Spearman's $r$-values are color-coded (blue to red); circle size indicates $p$-value. **d** Frequency of proviruses positive for various 2-, 3-, or 4-target combinations per $10^6$ CD4$^+$ T cells across two time points in 13 PWH. Each circle represents one readout; size indicates abundance of each target combination; color denotes the specific combination. Participant IDs are shown along the x-axis; samples from two time points are labeled (1, 2), except for participant 5101 (single time point available). For participant 5106 reservoir size was only quantifiable with IPDA-based Q4ddPCR at time point 1. Red squares highlight samples with $\Psi$ detection failure in one Q4ddPCR variant that was rescued by the alternative version (Extended Data Fig. 5). Orange squares highlight sequence-confirmed intact, but env-$\Psi$-negative proviruses from participant 9242 that declined over time. Q4-based Q4ddPCR results are shown. Source data are provided as a Source Data file.

In our data set, we identified samples (participants 9244, 9255) with sequence-confirmed intact proviruses lacking env-$\Psi$ signal due to mismatches in the $\Psi$-binding site (Fig. 3b, Supplementary Fig. 6). This was resolved in some cases using the alternative $\Psi$ primer/probe set; however, in participant 9244, both $\Psi$ sets failed despite the presence of intact sequences confirmed by Q4PCR. To account for such cases without resorting to custom primer design, we included env-gag and env-gag-pol combinations in the decision tree. Although these readouts may slightly overestimate reservoir size due to imperfect shearing correction, the impact is limited and outweighed by the improved sensitivity of including these combinations. We excluded env-pol combinations due to their lower positive and negative predictive value (Supplementary Table 5). Applying the decision tree enabled intact provirus quantification in all 13 participants, highlighting the enhanced robustness gained by assay modularity.

In summary, these results demonstrate that multi-target flexibility in Q4ddPCR allows for standardized and sequence-informed readout strategies that preserve high sensitivity while improving specificity across genetically heterogeneous HIV reservoirs.

## Q4ddPCR preserves IPDA information yet aligns more closely with inducible virus

To assess the performance of Q4ddPCR on clinical cohort samples, we applied Q4ddPCR to peripheral blood CD4$^+$ T cells from 27 ART-treated individuals with HIV-1 subtype B enrolled in the Berlin KOHIVI cohort (Table 1). All participants had undetectable plasma viral loads at sampling. We performed side-by-side comparisons among Q4-based Q4ddPCR, IPDA-based Q4ddPCR, and the original IPDA. For two individuals, limited sample availability precluded the IPDA-based Q4ddPCR; therefore, only IPDA and Q4-based Q4ddPCR were applied.

Across all samples, intact proviruses were quantifiable in 26 participants using Q4ddPCR, compared to 23 participants by IPDA. Only one participant had intact proviruses detected by IPDA but not by Q4ddPCR. Notably, all cases of IPDA failure were successfully resolved through alternative readouts or primer/probe sets available within the Q4ddPCR assay, underscoring its robustness and flexibility in accommodating sequence variability and resulting primer/probe mismatches (Fig. 4a, c).

Total HIV DNA levels measured by Q4ddPCR and IPDA were highly correlated and concordant (Spearman $r = 0.98$, $p < 0.0001$, $\rho_c = 0.92$ for Q4-based and $\rho_c = 0.96$ for IPDA-based Q4ddPCR, Fig. 4a, b). The env-$\Psi$ readout of Q4ddPCR, which parallels the IPDA readout, also correlated strongly with IPDA measurements, especially when using the IPDA-based Q4ddPCR design. This particular correlation was driven partly by matching signal dropout for the $\Psi$-target in 3 of 27 participants, as both assays failed to detect these proviruses. These $\Psi$-target failures in Q4ddPCR were rescued by switching to the second $\Psi$ probe (Fig. 4a).

In contrast, intact provirus measurements differed considerably between Q4ddPCR and IPDA, with ≥ 2-fold differences observed in 78% of samples using Q4-based Q4ddPCR and in 72% using IPDA-based Q4ddPCR (Fig. 4c). The geometric mean ratio of intact provirus frequencies measured by Q4ddPCR vs. IPDA was 0.40 (range 0.15-2.0) for the Q4-based assay and 0.38 (range 0.06-1.5) for the IPDA-based assay. These results are consistent with prior evidence suggesting that IPDA overestimates intact proviruses relative to multi-probe or sequence-based methods. Overall, our Q4ddPCR preserves the information provided by IPDA in terms of high concordance for total HIV DNA or env-$\Psi$ quantification but offers greater flexibility and improved accuracy in assessing the frequency of intact proviruses, particularly in the context of primer and/or probe mismatches.

To further evaluate the biological relevance of Q4ddPCR readouts, we examined their relationship with inducible, replication-competent virus measured by quantitative viral outgrowth assay (QVOA) in a subset of 16 ART-treated participants enrolled in different observational studies (Supplementary Table 3). Viral outgrowth was quantified by p24 enzyme-linked immunosorbent assay (ELISA) 14 days after stimulation.

Intact proviral frequencies measured by Q4ddPCR showed strong correlations with QVOA (Q4-based Q4ddPCR: Spearman $r = 0.74$, $p = 0.01$; IPDA-based Q4ddPCR: $r = 0.52$, $p = 0.02$), whereas IPDA intact counts did not correlate significantly with QVOA ($r = 0.34$, $p = 0.19$; Fig. 4d).

Q4ddPCR intact proviruses were determined using the decision tree (Fig. 3c): 11 of 16 samples yielded 4-target-positive proviruses, while the remaining samples were classified based on 3-target positives or, in a single case, 2-target positive proviruses. As expected, QVOA frequencies were lower than Q4ddPCR-derived intact provirus estimates, but the fold differences between QVOA and Q4ddPCR were smaller and more consistent than those observed for IPDA (Fig. 4e).

Together, these results indicate that Q4ddPCR closely reflects the inducible reservoir, supporting its enhanced specificity for identifying replication-competent proviruses through multi-target PCR.

## Q4ddPCR enables more sensitive and reliable detection of intact HIV proviruses than IPDA across longitudinal samples

To evaluate Q4ddPCR performance in assessing HIV reservoir dynamics, we applied the assay to 99 samples from 42 participants enrolled in the ACTG A5321 cohort. All participants had chronic HIV-1 subtype B infection, initiated ART in ACTG trials for treatment-naive persons, and achieved sustained virologic suppression (HIV-1 RNA < 50 copies/mL) by week 48 of ART, with no reported treatment interruptions (Fig. 5a, Table 2)[29]. Due to limited DNA availability per sample and to facilitate comparisons with existing IPDA data from later time points

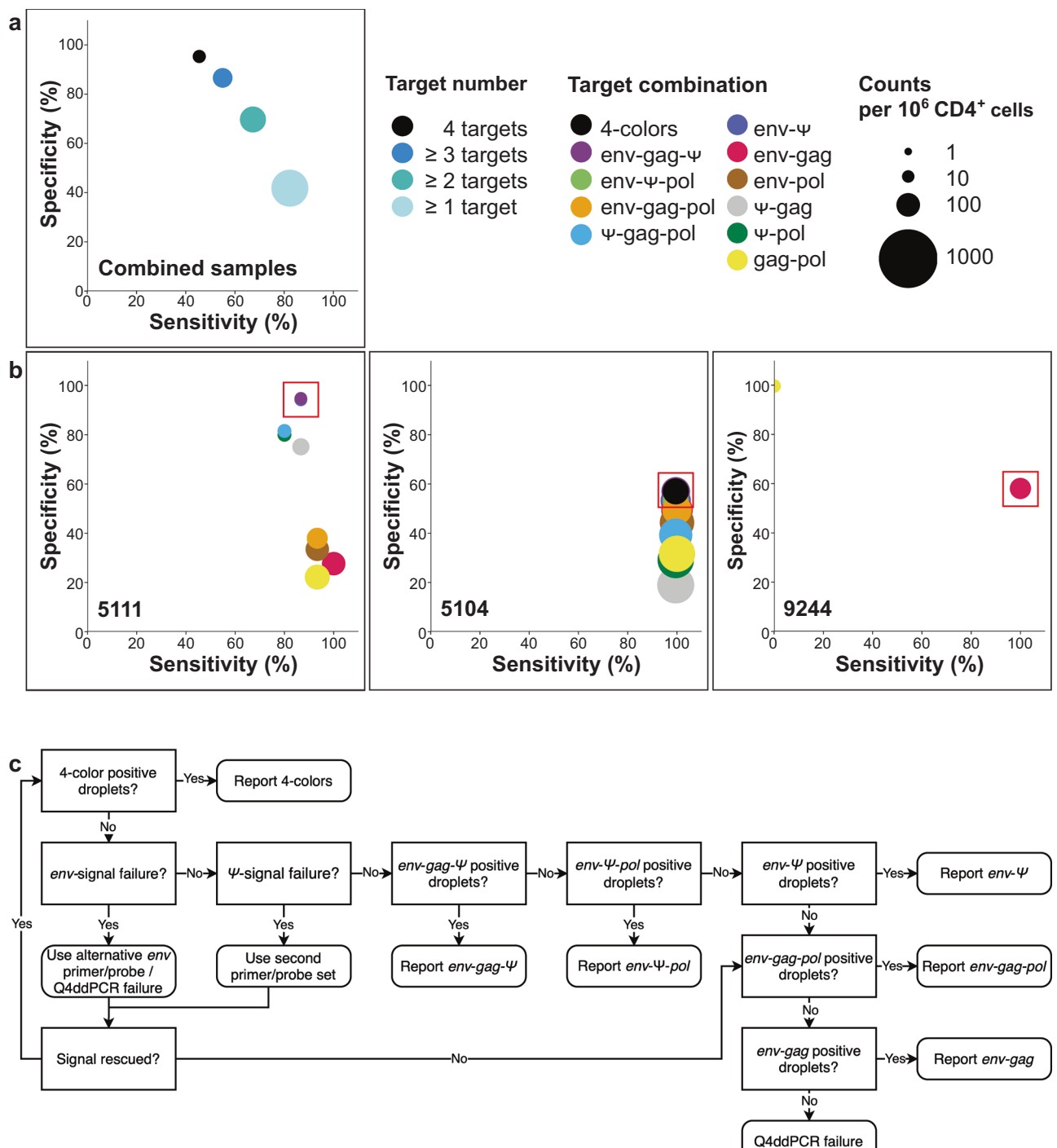

**Fig. 3 | Sensitivity and specificity of Q4ddPCR readouts for detecting intact proviruses. a**, **b** Sensitivity and specificity for intact provirus detection were evaluated across distinct Q4ddPCR target combinations using participant-matched Q4PCR-derived near full-length proviral sequences as a reference for 13 people with HIV (PWH). Sensitivity was calculated as the number of sequence-confirmed intact proviruses detected by a given target combination divided by the total number of intact sequences. Specificity was defined as the fraction of defective sequences not detected by the same target combination, relative to all defective sequences. Primer and probe sequences of Q4-based Q4ddPCR match those from Q4PCR. **a** shows pooled results from all participants (average); **b** shows data from individual

participants. Circles denote specific target combinations; size corresponds to the number of proviruses detected per 10⁶ CD4⁺ T cells, and color encodes the number (**a**) or specific combination (**b**) of amplified targets. Red squares mark the target combination selected by the decision tree. Results of Q4-based Q4ddPCR from the first available time point are shown. **c** Decision tree for reporting intact proviral frequencies from Q4ddPCR. The decision tree guides the selection of target combinations that maximize sensitivity and specificity while accounting for inter- and intra-individual proviral sequence heterogeneity. Source data are provided as a Source Data file.

in this cohort, we primarily used IPDA-based Q4ddPCR, supplemented by Q4-based Q4ddPCR in cases of amplification failure. Notably, *env* signal dropout was resolved in several cases by using a published backup primer/probe set[20].

Overall, Q4ddPCR successfully quantified intact proviruses in 94% (93/99) of samples, compared to only 77% (76/99) for IPDA. Signal dropout for at least one target occurred less frequently with Q4ddPCR (11%) compared to IPDA (15%). Importantly, Q4ddPCR identified

4-target positive proviruses in 64% of samples, and at least 3-target positive intact proviruses in 81% (Fig. 5b). Furthermore, 74% (17/23) of IPDA failures were rescued using alternative primer/probe sets (13/17) or readout combinations available within Q4ddPCR (*env-gag-pol* (1/17)

or *env-gag* readout (3/17)), across a wide dynamic range (7-722 intact proviruses per $10^6$ CD4$^+$ T cells) (Fig. 5d). These findings highlight that the increased sensitivity and robustness of Q4ddPCR are beneficial across a broad spectrum of reservoir sizes, not limited to samples with small reservoir sizes.

## Table 1 | KOHIVI Cohort Characteristics

| Number of participants | 27 (9 female, 18 male) |
|---|---|
| Age at sampling (years) | 55 (29–79) |
| Time on ART (years) | 12.9 (3.3–35.2) |
| CD4$^+$ cell count at sampling (/µL) | 720 (350–1850) |

Medians and ranges are indicated. µL, microliter.

### Q4ddPCR reveals a lower proportion and steeper decline of intact proviruses compared to IPDA

As anticipated, the proportion of proviruses detected varied based on the number and combination of Q4ddPCR-targets (Fig. 5c). Frequencies of intact proviruses differed substantially between Q4ddPCR and IPDA, with a geometric mean ratio Q4ddPCR/IPDA of 0.61 (range 0.03 – 1). Across all samples, Q4ddPCR reported significantly lower

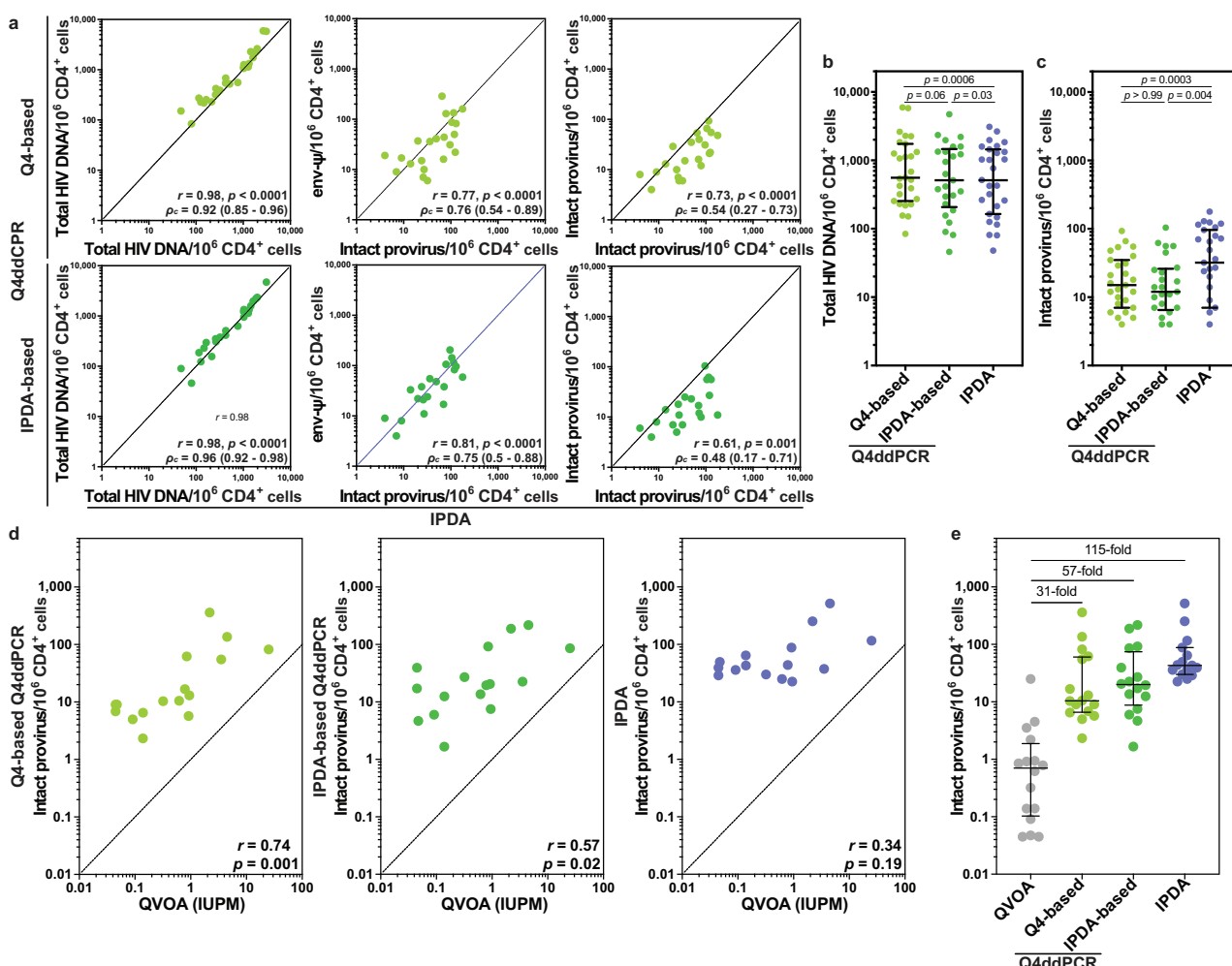

**Fig. 4 | Q4ddPCR preserves IPDA information but correlates with viral outgrowth. a–c** HIV reservoir size was assessed in parallel by Q4ddPCR and the intact proviral DNA assay (IPDA) in 27 antiretroviral therapy (ART)-treated participants from the Berlin KOHIVI cohort. **a** Correlation between reservoir measurements by Q4ddPCR (y-axis) and IPDA (x-axis). Left: total HIV DNA measurements by Q4-based (light green; top) or IPDA-based Q4ddPCR (dark green; bottom) vs. IPDA. Total HIV DNA was calculated as the sum of all proviruses positive for any single target, corrected for overlap due to proviruses harboring multiple targets. Q4ddPCR-measured total HIV DNA shows overall concordance with IPDA-derived values. Middle: *env-Ψ* readouts from Q4ddPCR correlate with IPDA-intact counts. Right: Intact proviruses measured by Q4ddPCR via the decision tree (Fig. 3c), plotted against IPDA-intact counts. Two-sided Spearman correlation (*r*) with *p*-values as well as concordance correlation coefficients (*ρ$_c$*) with 95% confidence intervals are shown for each panel. Side-by-side comparison of proviral frequencies measured by Q4-based (light green), IPDA-based (dark green) Q4ddPCR, and IPDA (blue) for

total HIV DNA (**b**) and intact proviruses (**c**). Each dot represents one sample (*n* = 27 for Q4-based Q4ddPCR and IPDA, *n* = 25 for IPDA-based Q4ddPCR). Medians with interquartile ranges are plotted. Statistical comparisons were performed using the two-sided Wilcoxon signed-rank test with correction for multiple comparisons. Samples with reservoir sizes measured as zero were included in all analyses but are not shown on the plots due to the use of a logarithmic axis, which cannot represent zero values. Reservoir size is expressed as proviruses per $10^6$ CD4$^+$ T cells. **d**, **e** Association between reservoir measurements by quantitative viral outgrowth assay (QVOA, grey) and Q4ddPCR (light and dark green) or IPDA (blue) in 16 ART-treated PWH. **d** Two-sided Spearman correlations are shown; QVOA correlated with Q4ddPCR but not IPDA. **e** Medians with interquartile ranges of QVOA, Q4ddPCR and IPDA results (*n* = 16 for QVOA, Q4- and IPDA-based Q4ddPCR, *n* = 15 for IPDA). Fold differences relative to QVOA are indicated. Source data are provided as a Source Data file.

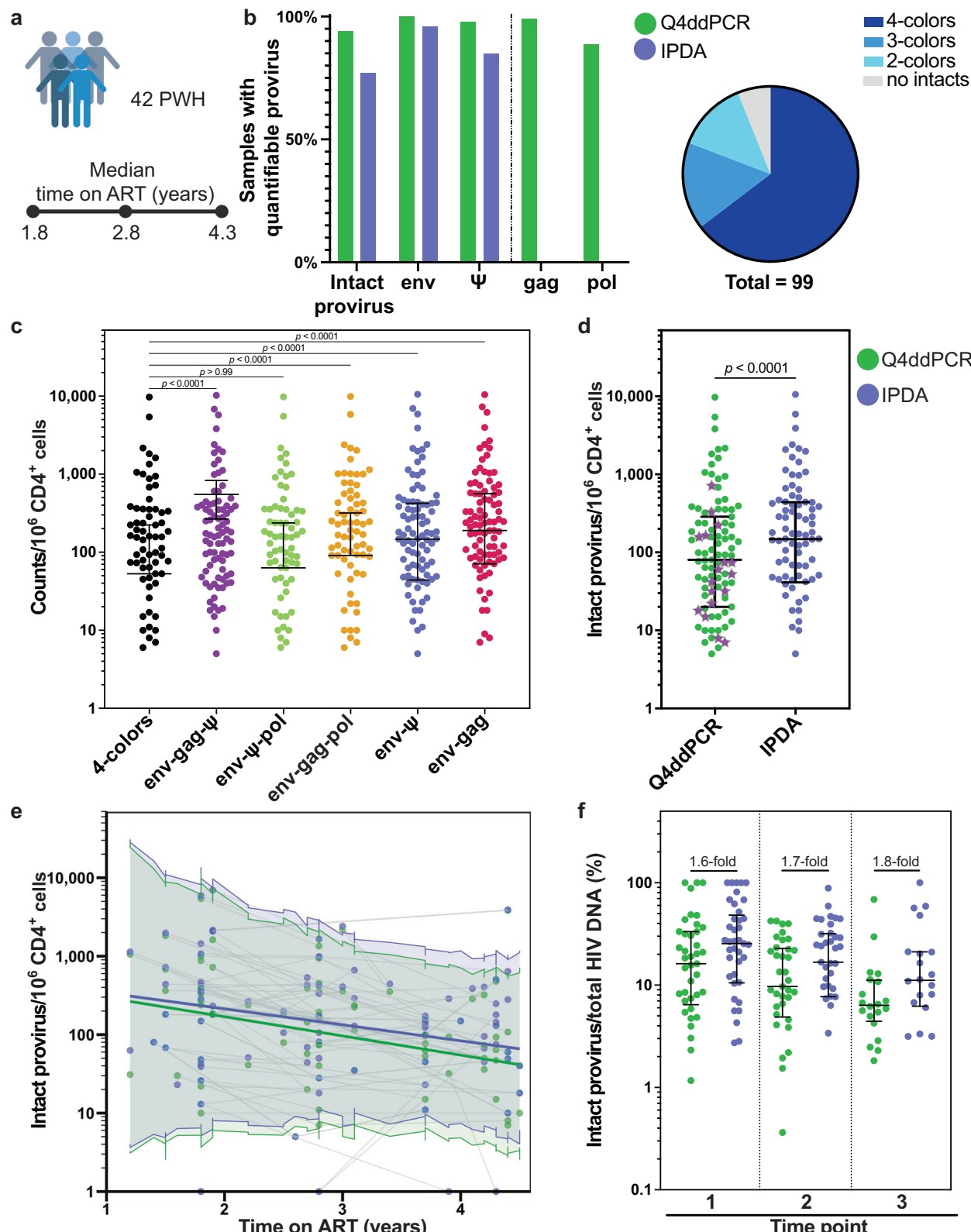

intact reservoir sizes than IPDA, underscoring its higher specificity (two-sided Wilcoxon signed-rank test, $p < 0.0001$, Fig. 5d).

Longitudinal analyses revealed a decline in intact proviral frequencies for most participants. Specifically, in 24 of 42 participants (57%), reservoir size decreased by at least 15% over a median follow-up of 2.3 years, although some participants exhibited increases or fluctuations. Decay rate modeling using a log-linear mixed-effects model

(accounting for random intercept and slope effects) showed consistent assay-dependent differences, although these did not reach statistical significance (Chi² test, $p = 0.33$). Q4ddPCR estimated a 43% annual decline (half-life 1.2 years, SE 0.3), compared to a 37% annual decline for IPDA (half-life 1.5 years, SE 0.4). The discrepancy between intact proviral frequencies measured by Q4ddPCR or IPDA widened progressively over time, becoming most pronounced at the latest time

**Fig. 5 | Q4ddPCR reveals a lower proportion of intact proviruses and a steeper decline over time compared to IPDA in longitudinal clinical samples. a** IPDA-based Q4ddPCR was applied to 99 longitudinal samples from 42 participants (33 male, 9 female) enrolled in the ACTG A5321 cohort. All participants had HIV-1 subtype B infection and initiated ART as part of ACTG clinical trials. Samples were collected within the first 4.5 years of suppressive ART. Median time on ART for each time point is indicated. Created in BioRender. Scheck, R. (2026) https://BioRender. com/34xgzfe. **b** Comparison of detection frequencies for intact proviruses and individual targets using Q4ddPCR (*env, Ψ, gag, pol*; green) and IPDA (*env, Ψ*; blue) across all samples (left). The right panel shows the distribution of samples based on the number of proviral targets simultaneously detected by Q4ddPCR to detect intact proviruses using the decision tree (2-4 targets; color coded). Intact proviruses were detected in 94% of all samples. **c** Intact proviral frequencies per $10^6$ CD4$^+$ T cells measured using different Q4ddPCR probe combination readouts (color coded). Dots represent samples; bars show medians and interquartile ranges. Statistical comparisons used the two-sided Friedman test with Dunn's correction. **d** Comparison of intact proviral frequencies measured by Q4ddPCR (green) versus IPDA (blue). Purple asterisks indicate samples where IPDA failed and Q4ddPCR

enabled quantification via alternative probe combinations and/or readouts. Each dot represents one sample. Medians and interquartile ranges are shown; two-sided Wilcoxon signed-rank test is used for statistical analysis. **e** Longitudinal decline of intact proviral frequencies over time as measured by Q4ddPCR (green) or IPDA (blue). Colored lines represent fixed-effect predictions from log-linear mixed-effects models, with shaded bands showing the 95% confidence intervals for model-predicted values. Each dot represents an individual sample; grey lines connect multiple samples from the same participant. **f** Ratio of intact proviral frequencies to total HIV DNA measured by Q4ddPCR (green) or IPDA (blue), stratified by time point (time point 1: n = 42, time point 2: n = 37, time point 3: n = 20). Fold difference between IPDA and Q4ddPCR is indicated. The greatest difference between methods is observed at later time points. Each dot represents one sample; bars show medians with interquartile ranges. Statistical comparisons between Q4ddPCR and IPDA were significant with a *p* < 0.0001 using the two-sided Wilcoxon signed-rank test. Samples with reservoir sizes measured as zero were included in all analyses but are not shown on the plots due to the use of a logarithmic axis, which cannot represent zero values. Source data are provided as a Source Data file.

## Table 2 | A5321 cohort characteristics

| | |
|---|---|
| **Number of samples** | **99** |
| **Number of participants** | 42 (9 female, 33 male) |
| ≥2 time points | 37 |
| 3 time points | 20 |
| **Age at baseline (years)** | 37 (21–63) |
| **Time on ART at baseline (years)** | 1.8 (0.9–4.3) |
| **Time on ART (years)** | 12.9 (3.3–35.2) |
| **CD4$^+$ cell count pre-ART (/μL)** | 296 (8–708) |
| **Viral load pre-ART (log$_{10}$ copies/mL)** | 4.7 (2.3–6) |

Medians and ranges are indicated. *μL* microliter. *mL* milliliter.

points (Fig. 5e). When excluding those samples where IPDA failed while Q4ddPCR rescued reservoir quantification, we obtained similar results, with an annual decline of 48% (half-life 1.1 years, SE 0.2) for Q4ddPCR and 39% (half-life 1.4 years, SE 0.3) for IPDA (Chi² test, *p* = 0.11).

Finally, Q4ddPCR revealed a consistently lower proportion of intact proviruses among total HIV DNA compared to IPDA with the highest relative difference observed at the latest time point. Specifically, the intact fraction measured by Q4ddPCR was 1.6-, 1.7- and 1.8-fold lower than by IPDA at the first, second, and third time points, respectively (Fig. 5f). Although the difference in estimated annual decay rates between assays did not reach statistical significance, this progressively widening difference experimentally validates mathematical modeling[7] and suggests that Q4ddPCR more accurately captures intact reservoir decay dynamics by more effectively excluding defective proviruses misclassified by IPDA - especially in scenarios where truly intact proviruses are preferentially being eliminated. These differences were not solely attributable to samples in which IPDA failed; rather, they reflect Q4ddPCR's enhanced ability to detect true reservoir changes. Thus, Q4ddPCR provides a more precise tool for longitudinally assessing intact HIV reservoir dynamics, potentially offering critical advantages for evaluating HIV cure interventions.

## Discussion

Reliable quantification of genetically intact HIV proviruses is essential for evaluating cure interventions[1]. In this study, we introduce Q4ddPCR, a 4-target droplet digital PCR assay that combines high specificity with throughput for intact proviral DNA detection, especially in clinical studies.

Definitive confirmation of genetic intactness would require sequencing at the level of individual droplets. Although microfluidic approaches show promise in this direction, they remain constrained by low throughput and technical limitations[33–35]. To overcome this, we

developed Q4ddPCR using biological samples previously characterized by extensive near full-length proviral sequencing, where the performance of each primer and probe had been directly matched to corresponding proviral sequence information. This design allowed us to rigorously benchmark assay performance against sequence-confirmed proviruses and confirm key predictions from previous mathematical modeling and Q4PCR studies[5,7,15].

We demonstrate that incorporating four probes significantly improves the accurate detection of intact proviruses as had been shown[5,7,15]. Although a fifth probe targeting the 5'LTR region is used in the related Rainbow assay[17], we decided not to include it on our ddPCR platform due to substantial fluorescence spillover into the *pol* channel. Importantly, we found that Q4ddPCR-derived total HIV DNA closely recapitulates the information provided by the 5'LTR assay, making the additional probe redundant.

Applying Q4ddPCR to samples from our pre-analyzed cohort, reservoir estimates derived from Q4ddPCR readouts showed strong correlation with sequence-confirmed Q4PCR, particularly its 4-target readout, suggesting that Q4ddPCR achieves a similar level of specificity, while offering enhanced scalability and simplicity through the ddPCR technique. However, while comparing Q4ddPCR with Q4PCR is advantageous as it provides the closest approximation to droplet-level sequencing, Q4PCR sequencing data is enriched for defective proviruses due to the inefficiency of long-distance PCR[26]. As a result, sensitivity and specificity estimates for Q4ddPCR may carry over certain inherent biases from the Q4PCR.

While an overall higher number of targets showed higher specificity, detailed participant-level analyses of specific target combinations revealed considerable variability, likely reflecting (1) viral sequence heterogeneity and (2) highly variable reservoir composition between individuals. To accommodate sequence heterogeneity, preserve inter-study comparability, and leverage semi-qualitative results from multi-probe readouts, Q4ddPCR has several strengths:

While primer/probe mismatches limit the performance of IPDA and related 2-probe assays to measure the intact reservoir[17,20,23], the multiprobe design of Q4ddPCR overcomes these limitations by using alternative readouts and therefore enables higher robustness towards sequence heterogeneity. In addition, the modular design of Q4ddPCR allows for backup primer/probe sets to recover signal when mismatches impair detection. Notably, in the ACTG A5321 cohort, 3 out of 5 samples with *env* amplification failures could be rescued using an alternative primer/probe set, with the limitation that this backup does not exclude hypermutated proviruses (contrarily to the original *env*-probe)[20].

To ensure standardization and interpretability of Q4ddPCR results across various clinical cohort studies, we developed a

sequence-informed, standardized reporting framework. Our decision tree prioritizes 4-color positive readouts as the highest-confidence metric, given their strong correlation with sequence-confirmed intact proviruses in Q4PCR and their superior specificity compared to standard IPDA. In samples lacking 4-target positives, 3-target combinations including both *env* and *Ψ* are prioritized, based on sensitivity and specificity considerations. As *env*-negative proviruses are most likely defective, we excluded them from the definition of intactness in our decision tree. In cases where *env-gag-Ψ* or *env-Ψ-pol* positives are absent, fallback to *env-Ψ* is recommended as it outperforms other 2-target combinations or *env-gag-pol*. However, certain participants, such as 9244, harbor sequence confirmed intact proviruses that are undetectable by *env-Ψ* but identifiable via alternative Q4ddPCR readouts like *env-gag*. These findings illustrate how Q4ddPCR expands detection sensitivity beyond conventional IPDA and related 2-target assays' limits. To accommodate such scenarios, our decision tree allows for fallback to alternative readouts after testing alternative *Ψ* primer/probe sets. While these fallback readouts are subject to slightly reduced specificity due to imperfect DNA shearing correction, their inclusion increases sensitivity without compromising interpretability, particularly in genetically diverse participant cohorts.

Overall, our observations corroborate *env-Ψ* as the most robust of all 2-target readouts but we see improved sensitivity when considering alternative readouts excluding *Ψ* as fallback, such as *env-gag-pol* and *env-gag*[6].

Collectively, out of the 179 samples from four distinct participant cohorts analyzed in this study, Q4ddPCR quantified intact reservoirs in 95% of cases, compared to only 79% by IPDA, highlighting the utility of this strategy across varying study populations. In the majority of samples (86%), intact reservoirs were quantified using the two highest hierarchies of our decision tree (4-colors and *env-gag-Ψ* readouts). Specifically, 65% of samples were quantified using 4-color readouts, 24% using 3-color readouts, and only 6% required 2-color readouts. This high rate of successful quantification is largely attributable to Q4ddPCR's modular, ready-to-use alternative probes and readouts, which distinguish it from previously published multi-probe decision-tree approaches[17].

The flexibility of the Q4ddPCR decision tree supports tracking of intact-reservoir dynamics across a wide range of reservoir sizes, including in participants with low-level reservoirs. A remaining limitation is that the current assay configuration has been validated only for subtype B. However, because of its modular architecture, Q4ddPCR can in principle be redesigned by substituting subtype-specific primer/probe sets (for example, published subtype C *env-* and *Ψ* primer/probes[21]), although this will require further experimental testing. Finally, our findings provide experimental support for a prior in silico prediction made during Q4PCR development that any two-target combination would detect 99% of HIV-1 subtype B samples[15]. This prediction held true in the ACTG cohort, with 98 out of 99 samples (99%) testing positive for at least one two-target combination.

To maintain compatibility with historical datasets and enable cross-study comparisons, we recommend reporting intact proviral frequencies applying IPDA-based Q4ddPCR. In the KOHIVI cohort, the Q4ddPCR *env-Ψ* readout closely matched IPDA estimates, showing that Q4ddPCR preserves IPDA information for studies seeking backward compatibility. At the same time, direct comparisons of intact proviral frequencies revealed that Q4ddPCR yields lower reservoir estimates than IPDA consistent with higher specificity.

Another key strength of Q4ddPCR lies in the availability of multilayered readouts that provide qualitative information and may enable tracking of the dynamics of individual proviral clones without sequencing. For example, in samples from participant 9242, we were able to see the dynamic of *env-Ψ*-negative, but sequence confirmed intact proviral clones with Q4ddPCR. These clones exhibited a clear decay trajectory, which would have been missed by IPDA alone.

Notably, recent studies have proposed that the change in clonality and clonal expansion of genetically intact proviruses contributes to the slowing or even reversal of reservoir decay during long-term ART[9,10]. These dynamics underscore the importance of tools that can monitor not only total reservoir size but also its composition over time. While sequencing remains the gold standard for clone tracking, Q4ddPCR's ability to capture distinct target combination patterns across time points offers a proxy readout for evolving reservoir clonality, giving quantitative and semiqualitative insights into reservoir composition.

An additional indication of the higher specificity of Q4ddPCR is its stronger correlation with functional reservoir measurements. Among a subset of 16 PWH with paired QVOA data, intact proviral frequencies measured by Q4ddPCR but not by IPDA significantly correlated with inducible replication-competent virus. This aligns with previous studies reporting moderate to weak or nonsignificant correlations between IPDA and QVOA[5,6,9,20,36], with improved correlations observed only after excluding samples with primer/probe mismatches or unreportable IPDA results[20,36]. In our dataset, participants were preselected for quantifiable IPDA results, allowing us to attribute the higher correlation to the increased specificity of Q4ddPCR rather than to the exclusion of mismatched or unreportable samples.

Mathematical modeling predicted that multi-probe measurements would more accurately capture intact reservoir decay dynamics by increasing specificity and minimizing the inclusion of slowly decaying defective proviruses[7]. In our longitudinal analysis of 42 PWH from an ACTG cohort, we observed a relatively rapid decline in intact proviral DNA during the early years on ART using Q4ddPCR. Furthermore, we estimated a median half-life of 1.2 years, compared to 1.5 years when using only the *env-Ψ*/IPDA readout. Although the difference in decay slopes did not reach statistical significance, the progressively widening gap between Q4ddPCR and IPDA in both intact proviral counts and the fraction of intact proviruses over time supports the interpretation that Q4ddPCR better resolves true reservoir decay. As predicted, assays that are more specific for intact proviruses will tend to estimate faster decays because they are more likely to exclude defective proviruses that are known to decay more slowly throughout long-term ART[7,13,14]. This may be particularly relevant for interventional studies where differences in intact reservoir decay between intervention arms are expected to be small, and the higher specificity of Q4ddPCR could improve the ability to detect biologically meaningful treatment effects.

The reservoir decline we observed in the A5321 cohort is faster than the half-lives reported in previous studies with PWH on ART[7,9,14,16]. However, our IPDA-derived half-life closely matches previously published estimates from individuals in the early stages of ART that estimated second-phase reservoir decay half-life of 1.6 years beginning approximately 3-4 months after ART initiation[8]. Our observations also align with findings from other analyses of our ACTG cohort. Previous work described a biphasic decay in HIV reservoir size, with a faster initial phase (approximately 1-year half-life) followed by a slower decline, with an inflection point occurring at a median of 5 years on ART[10]. Our participants, being at early ART time points (0.9 – 4.5 years), fall into this rapid decay window. The fluctuations we observed in individual participants are consistent with the interindividual variability previously reported in this cohort[10]. In conclusion, the improved specificity of Q4ddPCR better reflects reservoir dynamics and the decay of intact proviruses as predicted by mathematical modeling.

QVOA and sequencing-based assays offer high specificity for detecting intact HIV but are limited by cost, labor, and throughput. In light of the rapidly evolving landscape of HIV-cure and remission studies[37,38], Q4ddPCR offers a rapid, scalable, and cost-effective alternative for quantifying intact proviruses. Its 4-target, modular design accommodates sequence polymorphisms and supports alternative readouts, while its low sample input requirement makes the assay feasible for large clinical cohorts and sample-restricted groups such as children with HIV. Collectively, these features position Q4ddPCR as a

practical, versatile tool for both large-scale clinical trials and studies involving sample-constrained populations.

Looking ahead, leveraging machine learning to decode complex readout patterns may further enhance resolution and predictive power, guiding the prioritization of specific participants for more in-depth analyses such as single-cell analysis and phenotypic profiling of HIV-infected cells[33–35,39]. As HIV cure strategies move toward precision interventions, the balance of throughput, specificity, and flexibility offered by Q4ddPCR will be essential to bridge the gap between molecular measurements and clinically actionable endpoints.

## Methods

### Participants and ethics statement
We studied 110 participants with HIV-1 subtype B infection enrolled in either published trials (https://www.clinicaltrials.gov; NCT03526848 and NCT02825797; EudraCT: 2016-002803-25)[27,28] or in observational studies at one of the following sites: Charité Universitätsmedizin Berlin, Germany (KOHIVI study), Maple Leaf Medical Clinic, Toronto, Canada, or the NewYork-Presbyterian Hospital at Weill Cornell Medicine, United States. Apheresis from people without HIV was collected from the Gulf Coast Regional Blood Center, United States. Ethical approval to conduct this study was obtained from University of Toronto (reference number: 12-378) and the Weill Cornell Medicine Institutional Review Boards (reference number: 21-03023420), as well as the Ethics Committee of Charité Universitätsmedizin Berlin (reference number EA2/077/23). Participants enrolled in studies of the AIDS Clinical Trials Group (ACTG) initiated ART in ACTG trials and were subsequently followed in the A5321 cohort[29]. A5321 participants were recruited from 17 clinical research sites in the United States. Each clinical research site had the ACTG-approved A5321 protocol and consent form, and its relevant parental protocols and consent forms, approved by their local institutional review boards. PBMC samples from the A5321 cohort were obtained from the ACTG Specimen Repository through new work concept sheet NWCS 578. All participants provided written informed consent before participation, and the studies were conducted in accordance with Good Clinical Practice.

### Proviral sequence data
Proviral sequence data was obtained from Q4PCR as published[6,15]. For accuracy calculation, sensitivity was defined as the proportion of intact sequences identified by a given Q4PCR-target combination relative to the total number of sequence-confirmed intact proviruses. Specificity was calculated as the proportion of defective sequences not detected by a given readout, relative to the total number of defective sequences (Supplementary Table 5).

### Q4ddPCR and 5'LTR ddPCR
CD4$^+$ T cells were isolated from 5 to 50 × 10$^6$ cryopreserved PBMC samples using the CD4$^+$ T Cell Isolation Kit (Miltenyi Biotec) or Easy-Sep™ Human CD4$^+$ T Cell Isolation Kit (Stemcell Technologies) according to the manufacturer's instructions. Genomic DNA was then extracted using the DNeasy Blood and Tissue Kits for DNA Isolation (Qiagen) and eluted in 60 μL H$_2$O per 5 ×10$^6$ CD4$^+$ T cells lysed and immediately stored at −20 °C for up to 12 weeks. DNA quality and concentration was assessed using NanoDrop 1000 spectrophotometer or Qubit 4 Fluorometer with the dsDNA Quantification HS Kit (Thermo Fisher Scientific). DNA samples with concentrations above 100 ng/μL were diluted with H$_2$O.

Up to 750 ng of DNA per well was combined with 9.5 μL supermix for probes (no dUTP) (Bio-Rad) and the 2.5 μL of Q4ddPCR primer/probe mixes. These mixes included four fluorescently labeled internal hydrolysis probes, along with an unlabeled hypermutant-env probe. The primer and probe sequences were based on published sequences, with modifications to fluorophore labeling[5,15,20,31,32]. Fluorophore labeling, concentration of primer and probes, as well as cycling conditions have

been optimized and validated on genomic DNA from J-Lat 6.3 cells (NIH HIV Reagent Program)[40] to obtain the optimal separation between different clusters and reduce 'rain' in the ddPCR reaction.

The sequences and concentrations for each target are as follows: *env:* Primer (0.225 μM): forward AGTGGTGCAGAGAGAAAAAAGAGC, reverse GTCTGGCCTGTACCGTCAGC, probe (0.0625 μM): /5-VIC-CCTTGGGTTCTTGGGA-MGBNFQ, unlabeled hypermutant probe for discrimination of hypermutations within the target: CCTTAGGTTCT-TAGGAGC- MGBNFQ; backup-primer (0.9 μM): forward ACTATGGGCGCAGCGTC, reverse CCCCAGACTGTGAGTTGCA, backup-probe (0.25 μM): VIC-CTGGCCTGTACCGTCAG-MGBNFQ, Packaging Signal (Ψ): Q4-based primer (0.9 μM): forward TCT CTC GAC GCA GGA CTC, reverse TCT AGC CTC CGC TAG TCA AA, Q4-based probe (0.25 μM) /5Cy5/TT TGG CGT A/TAO/C TCA CCA GTC GCC /3IAbRQSp/, IPDA-based primer (0.9 μM): forward CAG-GACTCGGCTTGCTGAAG, reverse GCACCCATCTCTCTCCTTCTAGC, IPDA-based probe (0.25 μM): /56-FAM/TTTTGGCGTACTCACCAGT-MGBNFQ; *gag:* Primer (0.9 μM): forward ATG TTT TCA GCA TTA TCA GAA GGA, reverse TGC TTG ATG TCC CCC CAC T, Q4-based probe (0.25 μM): /5'6-FAM/CC ACC CCA C/ZEN/A AGA TTT AAA CAC CAT GCT AA/3IABkFQ/, IPDA-based probe (0.25 μM): /5Cy5/CCACCCCAC/TAO/AAGATTTAAACACCATGCTAA/3IAbRQSp/; *pol:* Primer (0.9 μM): forward GCA CTT TAA ATT TTC CCA TTA GTC CTA, reverse CAA ATT TCT ACT AAT GCT TTT ATT TTT TC, probe (0.25 μM): /5ATTO590N/AAGCCAGGAATGGATGGCC/3IAbRQSp/. For the 5'LTR ddPCR 11.5 μL of the supermix for probes (no dUTP) (Bio-Rad) was combined with the RU5 primer/probe mix: Primer (0.9 μM): forward TTA AGC CTC AAT AAA GCT TGCC, reverse GTT CGG GCG CCA CTG CTA GA, probe (0.25 μM): /56-ROXN/CC AGA GTC ACA CAA CAG ACG GGC ACA /3IAbRQSp/.

Primers and probes were purchased from Integrated DNA Technologies (IDT), except for the *env-*, unlabeled, and IPDA-based Ψ-probes, as well as probes needed for the RPP30 assay, which contained a minor groove binder (Thermo Fisher Scientific). Reactions were performed in a total volume of 20 μL per well, in up to 20 replicate wells. Droplets were generated with the manual or automated QX200 droplet generator (Bio-Rad). Thermocycling was performed with a 2 °C ramp rate, with an initial denaturation at 95 °C for 10 min, followed by 60 cycles (30 s at 94 °C and 1 min at 55 °C per cycle) and a final step 10 min at 98 °C before incubation at 4 °C. Droplets were read on the QX600 Droplet Reader (Bio-Rad). IPDA and parallel RPP30 assays for shearing correction and calculation of cell equivalents in IPDA and Q4ddPCR as previously described in four replicate wells per sample[5]. Wells containing fewer than 7,500 droplets or samples with less than 40,000 cell equivalents were excluded. We recommend aiming for 200,000 cell equivalents and considering differences in analyzed cell equivalents when evaluating longitudinal samples. Positive controls with J-Lat 6.3 DNA (Fig. 1b, c, Supplementary Fig. 1c) and negative controls using (Supplementary Fig. 1b, d) genomic DNA from people without HIV, as well as no-template controls, were performed in duplicates. Gates were manually set for each sample using QX Manager Software, Standard Edition v2.0. Merged replicates were gated in 2D scatter plots for each 2-target combination. Replicate wells were merged and gated in two-dimensional scatter plots for all possible 2-target combinations. In cases where merged replicates showed unclear cluster separation, replicates were inspected and gated individually. 'Cluster data' and 'data table' were exported and further processed using in-house R-codes (codes and usage documentation available on Buchauer lab and Gaebler lab GitHub page). These scripts automate the calculation of 1-, 2-, 3-, and 4-target-positive proviruses. Total HIV DNA is computed within the 'MultiplexPCRAnalyser' package as the sum of single-target signals corrected for overlapping shear-corrected target combinations. A sample-wise decision tree is applied across all available time points using the 'Processing_Q4ddPCR.R' workflow to minimize dropouts and ensure consistent classification.

## Quantitative viral outgrowth assays

For the quantitative viral outgrowth assay (QVOA) CD4$^+$ T cells were isolated from 100 to 300 ×10$^6$ PBMCs per participant using the EasySep™ Human CD4$^+$ T Cell Isolation Kit (Stemcell Technologies) and cultured in RPMI media (10% FBS, 1% Hepes, 2 mM L-Glutamine, 1% penicillin/streptomycin) with 50 U/mL IL-2 and 10 ng/mL IL-15. CD4$^+$ T cells were seeded in each 12 replicates of dilutions ranging from 0.05 ×10$^6$ to 1 ×10$^6$ CD4$^+$ T cells per well in 24-well plates and stimulated with 1 μg/mL phytohemagglutinin (PHA) and 1 ×10$^6$ irradiated feeder cells from PWoH. Total number of CD4$^+$ T cells used for QVOA was determined by Q4ddPCR results (total number of CD4$^+$ T cells used: 22.8 × 10$^6$ in 14 participants, 11.4 × 10$^6$ in 2 participants). After 24 hours media was changed to remove PHA, and 0.2 × 10$^6$ MOLT-4/CCR5 cells (NIH HIV Reagent Program)[41] were added to each well. Cultures were maintained for 2 weeks, changing media every three days. p24 enzyme-linked immunosorbent assays (ELISA) were run on supernatants collected at day 14 to identify positive wells. Absorbance values above the absorbance of the lowest p24 standard (3.25 pg/μL) and the average absorbance plus 3 standard deviations of negative controls were designated as p24-positive. The frequency of inducible cells was calculated through the IUPM algorithm developed by the Siliciano laboratory (http://silicianolab.johnshopkins.edu)[42]. We detected viral outgrowth in all samples tested.

## Statistical analyses and modeling of reservoir decay

To analyze reservoir decay dynamics, we used a log-linear mixed-effects model with fixed effects for time on ART, assay type (Q4ddPCR or IPDA), and their interaction, as well as participant-specific random effects for both intercept and slope. IPDA results with signal failures were excluded from analysis. Reservoir size values of 0 were set to 1. Model fitting was assessed by inspecting residual plots, Q-Q plots, and histogram distributions. Model convergence was verified via the optimization status code.

Statistical analyses were performed using GraphPad Prism Version 10.5.0 and R 4.5.0 for Mac OS X. For the calculation of the concordance correlation coefficient ($\rho_c$) reservoir values were log$_{10}$-transformed, and values of 0 were set to 1.

## Reporting summary

Further information on research design is available in the Nature Portfolio Reporting Summary linked to this article.

## Data availability

The data supporting the findings of this study are provided in the main figures, tables, and supplementary materials. Proviral sequence data have previously been published[5,7,15] and is available in GenBank with the accession numbers MN090187-MN090943, MT189273-MT191008, MT191115-MT191120, MW059111-MW059266, MW059441-MW059533, MW059602-MW059688, and MW060242-MW063069. Individual-level clinical data are subject to data protection regulations; therefore, only summary data are reported. Source data are provided with this paper.

## Code availability

Software to extract Q4ddPCR counts per 10$^6$ CD4$^+$ T cells from files exported from QX Manager Software can be found on the Buchauer lab GitHub page and has been archived on Zenodo (https://doi.org/10.5281/zenodo.15791354). The script used to run analyses with this R package as well as the code for the log-linear mixed effect model are available in a separate repository on the Gaebler lab GitHub page and have been archived on Zenodo (https://doi.org/10.5281/zenodo.16414846).

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

## Acknowledgements

We sincerely thank all study participants for their invaluable contribution to this research. We also express deepest appreciation to study teams and the participating study sites. We thank the processing lab of the Infectious Disease Department at Charité and all members of the R.B.J. and C.G. laboratory for discussions and support. We thank Sabine Weickmann for excellent technical assistance and Shy Genel for assistance with MATLAB coding. The ACTG study is funded by the National Institute of Allergy and Infectious Diseases (NIAID), grant numbers UM1 AI068634, UM1 AI068636, and UM1 AI106701. The following reagents were obtained through the NIH HIV Reagent Program, Division of AIDS, NIAID, NIH: MOLT-4/CCR5 Cells, ARP-4984 and J-Lat Full Length Cells (6.3), ARP-9846, contributed by Dr. Eric Verdin. R.S. is funded by the Deutsche Forschungsgemeinschaft (DFG, German Research Foundation, Walter Benjamin Fellowship, 549131684). D.B.R. acknowledges funding through the NIH grant R01 AI186721-01. C.G. was supported by the HJH-Foundation, and the Hector-Foundation. C.G. is a Charité-Foundation Recruiting Grantee and received support by the European Research Council (ERC) under the European Union's Horizon 2020 research and innovation program (grant agreement no. 101162138, Project "HIV CURE MISSION"). This work was also supported in part by the following NIH NIAID grants: UM1AI64565 (REACH Martin Delaney Collaboratory), UM1AI191237, R37AI181626, R01AI176601, R01AI170245, R01AI184285, and R01AI165301 (R.B.J). UM1AI164565 is also supported by NIMH, NIDA, NINDS, NIDDK, and NHLBI.

## Author contributions

R.S., A.R.W, R.B.J. and C.G. conceived, designed and coordinated experiments. R.S., G.G., L.L., N.P. and T.T.H. performed experiments, R.S. analyzed experiments, M.M. and L.B. wrote the R-code for analysis, D.K.M, R.J.B., B.J.M., J.C.C., J.J.E., J.W.M., R.T.G. collected clinical data, R.S. and D.B.R. performed and analyzed mathematical modeling, R.S. designed figures, R.S., R.B.J and C.G. wrote the manuscript with input from all co-authors.

## Funding

## Competing interests

R.B.J. has served as an advisor to ViiV Healthcare and received payment for this role. The remaining authors declare that they have no competing interests.
