## [Peer Review file · Nature Communications]

Q4ddPCR: A Flexible, 4-Target Assay for High-Resolution HIV Reservoir Profiling

Corresponding Author: Professor Christian Gaebler

Version 0:

Reviewer comments:

Reviewer #1

(Remarks to the Author)

Scheck and co-authors developed Q4ddPCR assay, an droplet digital PCR-based assay to measure intact (and defective) HIV proviral DNA. Taking advantage of the recent advances in digital PCR technology that allow to detect more than two colors at the same time, the assay simultaneously targets four conserved regions in the HIV genome. The primers and probes that are used in this assay are not new, but have been developed and published previously by the authors (Q4PCR assay) or others (IPDA assay). The principle of Q4ddPCR is the same as that of IPDA, the only difference is that four regions in the HIV genome are targeted instead of two. In this regard, q4ddPCR can be considered an improved version of IPDA. Not surprisingly, Q4ddPCR appears to be a more specific assay to measure the intact reservoir than IPDA, as the addition of two targets provides a better specificity for intact proviruses. Also not surprisingly, however, the sensitivity of the assay is decreased compared to two-target assays such as IPDA, as more targets mean more possibilities for false-negative results due to primer/probe-template mismatches. In particular, Q4ddPCR with four targets could detect only ~40% of sequence-confirmed intact proviruses, leading to the serious underestimation of the intact reservoir. The authors try to mitigate the sensitivity issue by using alternative readouts (two-target or three-target readouts, see decision tree at fig. 3c). However, the whole point of this assay was to add more targets in order to increase precision and specificity for intact proviruses. It is surely possible to use 2 or 3 targets instead of 4, which will increase sensitivity - but at the cost of specificity and precision.

Overall, the manuscript is well written and the data are clearly presented. However, this reviewer feels that the conceptual advance provided by this study is rather limited, for the reasons specified above and also because a very similar "multi-color" assay has been already described in a publication by Delporte et al. (Integrative Assessment of Total and Intact HIV-1 Reservoir by a 5-Region Multiplexed Rainbow DNA Digital PCR Assay, Clin Chem 71, 203-214, 2025). That publication reports a 5-region digital PCR, which is even one region more than the current manuscript so in theory should provide an even better specificity for intact proviruses. The only difference is that the current assay is based on QX600 droplet digital PCR system (Bio-Rad) and the "Rainbow assay" from Delporte et al - on QIAcuity Digital PCR System (Qiagen). But the principle is the same. Even the "decision tree" (fig. 3c in this manuscript) is not new as a very similar decision tree is present in the Delporte paper (fig. 6).

Minor point: it would be useful to provide an exact p value of the comparison between Q4PCR and 4-target Q4ddPCR in figure 2b. It only says "ns" (not significant) which means it is 0.05, but a p value of 0.5 is surely different from a p value of 0.06. From looking at the figure, it is clear though, that Q4ddPCR still overestimates the number of intact proviruses measured by Q4PCR.

(Remarks on code availability)

Reviewer #2

(Remarks to the Author)

Scheck et al report the modification of the sequencing based Q4PCR to a ddPCR based assay for high throughput quantification of intact proviruses. The assay is validated on a previously sequenced cohort and is then used to describe intact proviruses in a number of cohorts including a longitudinal cohort where they show a faster decline in intact proviruses compared to standard IPDA. There appears to be good separation between the different populations which is good and the

assay seems robust. Overall they show that Q4ddPCR reports less intact proviruses compared to standard IPDA which they attribute to IPDA overcalling intact proviruses. There is no doubt that correct enumeration of intact proviruses is of critical need to the field however, a number of other assays have been published using more than the standard two targets such as the 5T-IPDA (PMID: 33948574) and the Rainbow assay (PMID: 39749517), although it appears as though the dropout rate is much reduced here. Therefore, it's an important technical advance. It doesn't solve one of the major issues holding back the field which is applicability to diverse subtypes. Specific comments below

1. What proportion of samples can be enumerated using the 4 target system and how many must revert to 3 or 2 targets. This is an important piece of information that would allow for the reader to understand how significant an improvement this assay is. Perhaps this can be reported in the decision tree in figure 3 (for each cohort or combined)?
2. Not fully clear to this reviewer, are the authors advocating for Q4ddPCR or for the IDPA-Q4ddPCR? The longitudinal analysis appears done with IDPA-Q4ddPCR. Perhaps this can be made clear in the abstract.
3. The differences in intact provirus decline observed in ACTG cohort, is there statistical justification to demonstrate these are real differences? They should be reported as this is claimed in the abstract
4. Assume difference between Fig 2 and extended Fig 2 is one is Q4ddPCR and one is the IPDA-based Q4ddPCR. The labeling is exactly the same on both though, could this be made clearer?
5. Fig 2b: many data points missing for '4 targets', is it because no counts observed? Can this be reflected in the graph?
6. Line 180: If interpreted correctly, it would be worth stating that psi was restored with IPDA-version in 2 of 4 donors and 1 of 4 restored using Q4-version
7. Line 185: This seems an important point but is this data captured in fig 2d and ext fig 2c - the circles for env-Ψ are bigger than env-gag and env-gag-pol. Can this be demonstrated in a different way?
8. Line 300: One assumed because this is IPDA-Q4ddPCR this is due to Psi failure and use of env-gag droplets to quantify intact? Can authors confirm this was how samples were rescued?
9. Line 407: Is this statement correct? Was there not a significantly higher detection of intact proviruses by IPDA (figure 4c)?
10. Line 526: Is it possible to provide further information about how droplets were classified as 2, 3 or 4 target positive? Was this performed using the biorad software or required in house R code? This information may be helpful for other labs that may be interested in establishing the q4ddPCR and making decisions about what machines to purchase (maybe even in extended data).

(Remarks on code availability)

Reviewer #3

(Remarks to the Author)

Scheck et al. present a multiplexed digital PCR version that is based on their previously published Q4PCR workflow backed up with a solid evaluation against full-length sequences. I would advise the authors to consider the following.

1/ Consult the dMIQE2020 update guidelines and checklist for dPCR experiment reporting to inspect whether or not additional information could be provided.

2/ Assay performance characteristics and metrics could be extended to determine LOB, LOD, LOQ, resolution between clusters. Are there dilution series performed on J-Lats? If not, would advise to include this. If sufficient NTCs were already run a LOB calculation could also be included. At the end the authors evaluate specificity and sensitivity but this could be confusing as these also can refer to the more technical aspects of an assay.

3/ There is competition between the ENV-GAG and the GAG-POL, were any optimization approaches tried to further orthogonalize the position of the clusters?

4/ In terms of visualization, it would be helpful to color the true 4 color positive partitions in the 2D or 3D plots, so a visual inspection can be done of where these partitions lay. This can help in thresholding these out as the abovementioned competition could result in partitions that are close to the single positive cluster.

5/ Add comment on potential subtype performance of the Q4ddPCR (subtype B oriented) or highlight flexible redesign opportunities of these type of assays to accommodate other subtypes.

6/ On the Decision tree. How does it compare to other decision trees that are published? Would add section on this.

7/ Would the authors have data to support or hint towards the merit of adding more regions for intactness readouts and if there would be a limit in added value of performing for instance a 10 plex reaction. In other words, where is the limit?

8/ The authors mention 40K cells as minimal input. Could the authors elaborate on this threshold number or other QC for input? Also, if a 0 results is present, could they comment on the absence of signal not meaning absence of intact virus.

9/ The authors might consider to rename to Q4dPCR in stead of Q4ddPCR, referring to all digital PCR systems rather than only droplet-based systems.

10/ The evaluation of Q4ddPCR with Q4PCR generated viral sequence data could skew/bias the absolute % of sensitivity

and specificity as the same primer sets are used. A cautionary note could be added while interpreting these values.

(Remarks on code availability)

Version 1:

Reviewer comments:

Reviewer #1

(Remarks to the Author)

The authors provided a revised version of the manuscript and an extensive response to reviewers comments. While I appreciate their efforts, I still think their work provides only an incremental technical advance over the state of the art (e.g. the Rainbow assay) for measuring the HIV reservoir, rather than any conceptual advance. While this assay will probably be useful for HIV reservoir researchers, this is essentially a technical report describing a variation of already published assays with the same underlying principle.

(Remarks on code availability)

Reviewer #2

(Remarks to the Author)

The authors have addressed my concerns and suggestions appropriately. The updated figures are improved for clarity. I have no further concerns.
Michael Roche.

(Remarks on code availability)

Reviewer #3

(Remarks to the Author)

No further comments.

I thank the team and authors for adding clarifications and desired adaptations to the manuscript.

(Remarks on code availability)

Response to Reviewers' comments

We would like to thank the reviewers for the time and effort they have dedicated to evaluating our manuscript and for the valuable feedback. The reviewers' suggestions have improved the technical depth of our analyses and enhanced the overall quality of the work, including the addition of new comparative data with functional reservoir analyses. Replies to the reviewers' comments (in green) are outlined below (in black), and all changes can be found in the revised manuscript version with tracked changes.

Reviewer #1 (Remarks to the Author):

We thank the reviewer for pointing out that:

“Overall, the manuscript is well written and the data are clearly presented.”

The reviewer raises the following points:

“Scheck and co-authors developed Q4ddPCR assay, a droplet digital PCR-based assay to measure intact (and defective) HIV proviral DNA. Taking advantage of the recent advances in digital PCR technology that allow to detect more than two colors at the same time, the assay simultaneously targets four conserved regions in the HIV genome. The primers and probes that are used in this assay are not new, but have been developed and published previously by the authors (Q4PCR assay) or others (IPDA assay). The principle of Q4ddPCR is the same as that of IPDA, the only difference is that four regions in the HIV genome are targeted instead of two. In this regard, q4ddPCR can be considered an improved version of IPDA. Not surprisingly, Q4ddPCR appears to be a more specific assay to measure the intact reservoir than IPDA, as the addition of two targets provides a better specificity for intact proviruses. Also not surprisingly, however, the sensitivity of the assay is decreased compared to two-target assays such as IPDA, as more targets mean more possibilities for false-negative results due to primer/probe-template mismatches. In particular, Q4ddPCR with four targets could detect only ~40% of sequence-confirmed intact proviruses, leading to the serious underestimation of the intact reservoir. The authors try to mitigate the sensitivity issue by using alternative readouts (two-target or three-target readouts, see decision tree at fig. 3c). However, the whole point of this assay was to add more targets in order to increase precision and specificity for intact proviruses. It is surely possible to use 2 or 3 targets instead of 4, which will increase sensitivity - but at the cost of specificity and precision.

We thank the reviewer for bringing this up and agree that the primary aim of Q4ddPCR is to maximize precision and specificity through the addition of more targets. In fact, 65% of all tested samples across 4 cohorts and 110 individuals yielded detectable 4-color positive proviruses, which provide the highest predictive value for intactness and the greatest potential to capture dynamic changes in the intact reservoir.

The relatively low fraction of sequence-confirmed proviruses showing Q4PCR 4-color positivity (45.5%, Extended Data Table 5) in our sequence-validation cohort reflects the intentional selection of challenging participants. This cohort was shown to have an unusually high signal failure rate in a thorough comparative validation study across different patient cohorts (Gaebler, Falcinelli et al. - JVI 2021, DOI: 10.1128/JVI.01986-20) and included individuals with known primer/probe mismatches (5106, 5114, 5115, 9242, 9243, 9244, 9255), known amplification failures (5114, 5115, 9243, 9244, 9255), very small reservoir sizes (5106, 5115) and unusually steep reservoir declines (5104). This cohort was chosen to rigorously validate Q4ddPCR with a high benchmark. Our goal was to assess whether a high-throughput platform such as Q4ddPCR could (1) capture this level of biological complexity and (2) provide practicable solutions such as validated ready-made backup primer/probes or alternative readouts to mitigate failures without compromising data interpretability. To this end, we used this cohort to implement systematic readouts guided by the 3,650 proviral sequences (derived from PCR signals using the same primer/probes as Q4ddPCR) and to develop the hierarchical decision framework that maximizes specificity across participants while preserving sensitivity.

The resulting 4-color readout in 65% of all tested samples as well as the very low drop-out rate of 5% across 179 samples demonstrates the success of this strategy and underscores the robustness of the Q4ddPCR framework. In addition, the strong correlation of decision-tree based Q4ddPCR results with functional viral outgrowth reservoir estimates underscores the precision and specificity of our approach.

[Redacted]

Similar issues with high drop-out rates have been observed in other trials for example, AELIX-003 which reported no statistically significant reservoir reductions but had high assay failure rates (20%: 10% in intervention and 40% in placebo group).

[Redacted]

The reviewer raises additional points:

“Overall, the manuscript is well written and the data are clearly presented. However, this reviewer feels that the conceptual advance provided by this study is rather limited, for the reasons specified above and also because a very similar "multi-color" assay has been already described in a publication by Delporte et al. (Integrative Assessment of Total and Intact HIV-1 Reservoir by a 5-Region Multiplexed Rainbow DNA Digital PCR Assay, Clin Chem 71, 203-214, 2025). That publication reports a 5-region digital PCR, which is even one region more than the current manuscript so in theory should provide an even better specificity for intact proviruses. The only difference is that the current assay is based on QX600 droplet digital PCR system (Bio-Rad) and the "Rainbow assay" from Delporte et al - on QIAcuity Digital PCR System (Qiagen). But the principle is the same. Even the "decision tree" (fig. 3c in this manuscript) is not new as a very similar decision tree is present in the Delporte paper (fig. 6).”

We thank the reviewer and agree that both Q4ddPCR and the Rainbow Assay rely on the concept of prioritizing high-specificity multi-color readouts as the strongest evidence of intactness. However, the use of the Bio-Rad ddPCR platform is a critical feature of Q4ddPCR since the current gold-standard IPDA assay uses the same platform. Q4ddPCR was intentionally designed to ensure comparability within the field as well as backwards compatibility with IPDA results.

In addition, our manuscript offers a deeper sequence-informed design and validation of the assay, together with evidence of successful implementation across multiple independent cohorts. As a result, Q4ddPCR differs from the Rainbow assay in several key respects:

First, Q4ddPCR incorporates ready-to-use backup primer/probe sets and alternative readouts that directly address IPDA amplification failures, reducing the overall failure rate from 12-30% to only 5%. In contrast, all alternative Rainbow-readouts still depend on *env-Ψ*-positivity and therefore do not mitigate these failures. Moreover, the alternative readouts of Q4ddPCR allow sensitive tracking of very small reservoir sizes when no 4-target positive proviruses are detected. By comparison, the Rainbow assay first tests for amplification failures at the single-channel level and only falls back to 4- to 2-color *env-Ψ*-positive readouts when failures are detected. In practical terms, this strategy leads to (1) underestimation of intact reservoirs in participants who do not have signal failures, but whose reservoir size falls below the limit of detection of a five-target assay and (2) high frequencies of samples with no reportable intact proviral values. Indeed, across the Rainbow cohort using their decision tree including the individualized primer/probes, intact reservoirs were quantifiable in only 59% of participants.

Second, the pre-validated alternative primer/probes and readouts in Q4ddPCR make it markedly more labor-efficient, cost-effective and suitable for large clinical trials than the Rainbow assay, which suggests designing individualized primer/probe sets. Our approach was driven by our experience in the eCLEAR trial (Gunst et al. – Nature Medicine 2023, DOI: 10.1038/s41591-022-02023-7), where we designed participant specific primer/probes for participants with IPDA failures which exceeded the capacities for most clinical trials. Avoiding individualized primer probe sets and decreasing the dropout rate in clinical trial is particularly important as it can compromise statistical power.

Therefore, we strongly believe that Q4ddPCR provides a flexible, scalable and specific tool that has major advantages over Rainbow assay and can be readily implemented in the field. In fact, Q4ddPCR is already being used in several clinical trials such as the ACTG A5386 trial (NCT04340596) as well as the landmark RIO study (NCT04319367).

We thank the reviewer for suggesting the following point:

“Minor point: it would be useful to provide an exact p value of the comparison between Q4PCR and 4-target Q4ddPCR in figure 2b. It only says "ns" (not significant) which means it is 0.05, but a p value of 0.5 is surely different from a p value of 0.06. From looking at the figure, it is clear though, that Q4ddPCR still overestimates the number of intact proviruses measured by Q4PCR.”

We thank the reviewer for this remark. The figure has been updated accordingly. The 4-target Q4ddPCR readout shows slightly higher intact proviral frequencies than Q4PCR, although the difference is not statistically significant. This pattern aligns with known limitations of Q4PCR, where the long-distance PCR step can underestimate full-length intact genomes by preferentially amplifying shorter defective proviruses. While some degree of overestimation by Q4ddPCR cannot be ruled out, underestimation by Q4PCR likely contributes to the observed difference between the assays.

Reviewer #2 (Remarks to the Author):

We thank the reviewer for pointing out that:

“Scheck et al report the modification of the sequencing based Q4PCR to a ddPCR based assay for high throughput quantification of intact proviruses. The assay is validated on a previously sequenced cohort and is then used to describe intact proviruses in a number of cohorts including a longitudinal cohort where they show a faster decline in intact proviruses compared to standard IPDA. There appears to be good separation between the different populations which is good and the assay seems robust.”

And that:

“Overall they show that Q4ddPCR reports less intact proviruses compared to standard IPDA which they attribute to IPDA overcalling intact proviruses. There is no doubt that correct enumeration of intact proviruses is of critical need to the field however, a number of other assays have been published using more than the standard two targets such as the 5T-IPDA (PMID: 33948574) and the Rainbow assay (PMID: 39749517), although it appears as though the dropout rate is much reduced here. Therefore, its an important technical advance.”

We thank the reviewer for recognizing the thorough sequence-based validation of Q4ddPCR and for noting that Q4ddPCR represents a meaningful technical advance.

In addition to a reduced assay dropout rate, we would like to highlight important differences between Q4ddPCR and other multi-target assays such as the Rainbow and 5T-IPDA assay:

	Q4ddPCR	Rainbow	5T-IPDA
Assay design	2 separate sets of primer/probes with 4 targeting regions (Q4- and IPDA-based Q4ddPCR): env , Ψ gag , and pol , one fluophore per target, primer/probes derived entirely from Q4PCR or env - and Ψ -primer/probes derived from IPDA, gag and pol derived from Q4PCR	5 target assay: env , Ψ , gag , pol and RU5 for total HIV DNA, one fluophore per target, computational compensation for signal spillover, env - and Ψ -primer/probes derived from IPDA, gag and pol derived from Q4PCR	5 target assay: Combination of 2 triplex ddPCRs with each two unique targets (3' end pol and tat , and LTR/ gag and 5'end pol) and one common target for inter-assay control (env -target), shared fluophores with different concentrations (FAM low and high, and HEX low and high), newly designed primer/probes

Reporting for intactness	Standardized decision tree, prioritizes specificity, includes ready-to-use backup primer/probes and alternative readouts	Standardized decision tree, prioritizes specificity, alternative readouts only in cases of amplification failure, only env-ψ-positive readouts , proposed backup strategy: design of custom primer/probes	5 targets must be positive for intactness
Drop-out rate	5% in 179 samples	41% in 83 samples	20% in 157 samples
Validation	Standard material and Q4PCR-target combination-matched sequences (3,650 sequences, 558 intact)	Standard material, HIV-PULSE sequencing and subsequent in silico analysis with defined criteria for mismatches (855 sequences, 29 intact)	Standard material, PBMCs infected in vitro with 6 HIV-1 isolates, in silico analysis against clade B sequences with defined criteria for mismatches (1071 sequences, 67 intact)
Criteria for dPCR assay performance	Target-combination matched qPCR signal in Q4PCR	In silico criteria: 5 mismatches in primers/probe set and a maximum of 3 mismatches per primer/probe	In silico criteria: 5 mismatches from the primer sequences and 0 mismatches from the probe
Additional specificity assessment	Correlation with QVOA (Q4-based assay Spearman's $r = 0.74$, $p = 0.001$ and IPDA-based assay $r = 0.57$, $p = 0.02$)	No data reported	Correlation with QVOA (Spearman's $r = 0.48$, $p = 0.02$)
Analysis	Gating in QX-Manager, automated processing with R, Output format: copies/10⁶ CD4⁺ cells shear corrected	Gating with ddpcRquant with manual adaptation, analysis with R Shiny web application (https://digpcr.ugent.be/software.html), Output format copies/μL	Automated gating, analysis either with Excel tables or R-script
Normalization and shear correction	RPP30	RPP30	Quantification per T cells (T cell receptor D gene) and RPP30
Platform	QX600 (Bio-Rad)	QIAcuity (Qiagen)	QX200 (Bio-Rad)

A key distinction between Q4ddPCR and other recently published multi-target assays such as the 5T-IPDA and Rainbow assay lies in the strategy used to evaluate assay performance. Both the 5T-IPDA and Rainbow assay validated assay specificity primarily through *in silico* matching of primers and probes to participant-derived proviral sequences. This approach is informative, yet it is also limited, because *in silico* predictions cannot always reliably determine whether a given pattern of mismatches will extinguish or merely attenuate PCR signal. As a result, *in silico* validation risks overestimating specificity by assuming that sequences predicted to fail amplification indeed fail in wet-lab conditions.

To avoid this limitation, we used a different validation strategy: we selected samples from 13 PWH for whom we had experimentally derived proviral sequences directly linked to Q4PCR signals generated with the same primer/probe sets used in Q4ddPCR. This allowed us to evaluate specificity using observed rather than predicted amplification behavior and provided a rigorous benchmark for developing the decision tree. The trade-off of this approach is that long-distance PCR used in Q4PCR under-amplifies full-length intact proviruses, likely causing a slight underestimation of Q4ddPCR specificity in our dataset. Besides validating Q4ddPCR, this also

provides valuable insights into reservoir composition of different individuals and shows that these insights can be achieved using high throughput methods as Q4ddPCR.

Notably, our evaluation included a much higher number of sequences, especially we used substantially more intact sequences (558) compared to 29 in the Rainbow assay or 67 for the 5T-IPDA, which allowed higher confidence in our specificity estimates.

Finally, Q4ddPCR was intentionally built and validated to be backward-compatible with IPDA, enabling integration into longitudinal cohorts where IPDA data already exist and ensuring continuity across ongoing studies. This differentiates Q4ddPCR from the Rainbow assay and 5T-IPDA and supports its practical adoption in the field.

The reviewer raises additional points:

“It doesn’t solve one of the major issues holding back the field which is applicability to diverse subtypes.”

We agree that applicability across diverse HIV subtypes remains a major challenge for all current molecular reservoir assays, including our own. Current implementation performs best for subtype B. However, because Q4ddPCR is modular, primer/probe sets can be exchanged by subtype-specific alternatives and our preliminary work with subtype C *env*- and Ψ -primer/probes demonstrates feasibility. Nevertheless, full validation across diverse subtypes will require future work.

“1. What proportion of samples can be enumerated using the 4 target system and how many must revert to 3 or 2 targets. This is an important piece of information that would allow for the reader to understand how significant an improvement this assay is. Perhaps this can be reported in the decision tree in figure 3 (for each cohort or combined)?”

Thank you for pointing this out. We agree that this is important information and have added text to the discussion of our manuscript. In our dataset of 179 samples, in 65% proviruses were detectable with 4 targets, while 24% used 3-target-positive proviruses as surrogate for intact proviruses (21% *env-gag- Ψ* , 2% *env- Ψ -pol*, 1% *env-gag-pol*). An additional 6% had to rely on 2-target-positive proviruses as readout for intactness (3% *env- Ψ* , 3% *env-gag*). In 5% of samples no intact proviruses were detected. Thus, a total of 86% of samples had a quantifiable reservoir using the two highest hierarchies of the decision tree.

“2. Not fully clear to this reviewer, are the authors advocating for Q4ddPCR or for the IDPA-Q4ddPCR? The longitudinal analysis appears done with IDPA-Q4ddPCR. Perhaps this can be made clear in the abstract.”

We thank the reviewer for this comment. To clarify both the Q4-based Q4ddPCR and the IPDA-based Q4ddPCR are implementations of the same 4-target Q4ddPCR framework. We validated the assay primarily using the Q4-based Q4ddPCR, as it employs exactly the same primer/probe sets as Q4PCR and therefore allows a direct, sequence-matched assessment of sensitivity and specificity.

We introduced the IPDA-based Q4ddPCR for two reasons: (1) to provide a practical and easy solution for Ψ -target failures observed in a subset of samples, and (2) to ensure backward compatibility with existing IPDA datasets. Because the A5321 cohort analysis was part of a longitudinal study for which IPDA measurements had been generated at different time points in the study, we used the IPDA-based Q4ddPCR for that longitudinal assessment, complementing it with Q4-based Q4ddPCR when amplification failures occurred. Overall, we observed similar performance and high correlation between our two Q4ddPCR assays as well as with viral outgrowth. For clarification, we have revised the abstract to explicitly introduce the two primer/probe sets.

“3. The differences in intact provirus decline observed in ACTG cohort, is there statistical justification to demonstrate these are real differences? They should be reported as this is claimed in the abstract”

We thank the reviewer for raising this important point. We performed a log-linear mixed-effects analysis to test whether the decline in intact proviruses observed with Q4ddPCR differed from that measured by IPDA. The difference in slopes between Q4ddPCR and IPDA was not statistically significant (Chi² test, $p = 0.33$). When excluding samples with IPDA amplification failures, the difference approached but did not reach statistical significance ($p = 0.11$). We have added this information and the p -values to the manuscript. Although the difference is not statistically significant, we believe the observed divergence is still meaningful to report for two reasons. First, it is consistent with prior mathematical modeling and with intact reservoir decline patterns observed using Q4PCR and second the separation between assays increases over time for both the absolute number of intact proviruses and the intact proviral fraction, suggesting a biological rather than stochastic difference.

“4. Assume difference between Fig 2 and extended Fig 2 is one is Q4ddPCR and one is the IPDA-based Q4ddPCR. The labeling is exactly the same on both though, could this be made clearer?”

Thank you, we added clearer labels to the figures.

“5. Fig 2b: many data points missing for ‘4 targets’, is it because no counts observed? Can this be reflected in the graph?”

The absence of data points in the “4-target” category is indeed due to samples in which no 4-color-positive proviruses were detected. This is due to our validation cohort which was selected to represent a high benchmark specifically including participants with known primer/probe mismatches and amplification-failures. This choice inherently increases the proportion of samples lacking 4-target (and to a lesser extent 3-target) positive counts.

To make this clearer, we have now explicitly added the missing data points to Fig. 2b and Extended Data Fig. 5a, allowing readers to directly visualize the samples in which no 4- or 3-target positives were observed.

“6. Line 180: If interpreted correctly, it would be worth stating that psi was restored with IPDA-version in 2 of 4 donors and 1 of 4 restored using Q4-version “

Thank you, we clarified this in the manuscript.

“7. Line 185: This seems an important point but is this data captured in fig 2d and ext fig 2c - the circles for env- Ψ are bigger than env-gag and env-gag-pol. Can this be demonstrated in a different way?”

We thank the reviewer for pointing out the importance of this observation. We clarified in the revised manuscript that in participant 9242 the dominant intact clone is env- Ψ -positive, but that a smaller population of sequence-confirmed intact env-gag- (and env-gag-pol-)positive clones is also present. These smaller intact clones decline over time and would be missed if only env- Ψ /IPDA-type readouts were used. This illustrates that Q4ddPCR may offer semi-qualitative insights into reservoir composition and clonal dynamics in a high throughput setting.

To address the reviewer’s suggestion for clearer visualization, we added a new panel to Extended Data Fig. 5. In this panel, we aggregated env- Ψ -positive and env-gag-positive signals separately. For direct comparison and visualization, we normalized Q4PCR sequence counts and Q4ddPCR proviral counts to the env- Ψ -value at the first time point. This allows direct comparison of the longitudinal dynamics captured by Q4PCR and Q4ddPCR and illustrates how Q4ddPCR detects changes in minor clones that would otherwise be overlooked.

“8. Line 300: One assumed because this is IPDA-Q4ddPCR this is due to Psi failure and use of env-gag droplets to quantify intact? Can authors confirm this was how samples were rescued?”

Yes, thank you for pointing this out. We added a statement on how they were rescued in the results text.

“9. Line 407: Is this statement correct? Was there not a significantly higher detection of intact proviruses by IPDA (figure 4c)?”

The comparison shown in Figure 4c compares intact proviral estimates obtained using the full Q4ddPCR decision tree with the conventional IPDA intact proviruses obtained in a separate reaction. Because the decision tree prioritizes 4- and some 3-target-positive classifications over the *env-Ψ*-readout, Q4ddPCR yields lower intact counts than IPDA in this comparison.

However, the original statement in the manuscript “*In the KOHIVI cohort, Q4ddPCR env-Ψ values closely matched IPDA estimates, showing that Q4ddPCR preserves IPDA information for studies seeking backward compatibility.*” refers to the middle panels of Figure 4a, where we directly compared IPDA intact proviral counts with the *env-Ψ*-only readout extracted from Q4ddPCR (i.e., intentionally ignoring *gag* and *pol* targets to mimic IPDA classification). In this comparison, Q4ddPCR and IPDA show high correlation and concordance (Spearman’s $r = 0.77$ and 0.81 , $p < 0.0001$; $\rho_c = 0.76$ [95% CI 0.54 - 0.89] and 0.75 [95% CI 0.50 - 0.88]).

This supports our conclusion that Q4ddPCR preserves IPDA-equivalent information, enabling backward compatibility for studies or cohorts with previously acquired IPDA time points while providing a more specific classification when the full decision tree is used.

“10. Line 526: Is it possible to provide further information about how droplets were classified as 2, 3 or 4 target positive? Was this performed using the biorad software or required in house R code? This information may be helpful for other labs that may be interested in establishing the q4ddPCR and making decisions about what machines to purchase (maybe even in extended data).”

We thank the reviewer for this helpful suggestion. Additional methodological details have been added to the revised manuscript. In brief: Initial droplet gating is performed in Bio-Rad’s QX Manager Software, using all six possible 2-dimensional fluorophore combinations. The gated data are exported and processed using an R package available on the Buchauer Lab GitHub (<https://doi.org/10.5281/zenodo.15791354>). To simplify use, the Gabler Lab GitHub page (<https://doi.org/10.5281/zenodo.16414846>) provides a guide, a script (*RunFile.R*) for running the package and an additional script (*Processing_Q4ddPCR.R*) that automates downstream processing. The ‘*RunFile.R*’ computes all possible 1-, 2-, 3-, and 4-target combinations, provides their concentrations (copies/μL) and normalized values (per million CD4⁺ T cells). It automatically applies shear correction and calculates total HIV DNA. The ‘*Processing_Q4ddPCR.R*’ script implements the Q4ddPCR decision tree to determine intact classification, a quality control, and if desired IPDA-equivalent values.

Reviewer #3 (Remarks to the Author):

We thank the reviewer for pointing out that:

“Scheck et al. present a multiplexed digital PCR version that is based on their previously published Q4PCR workflow backed up with a solid evaluation against full-length sequences.”

The reviewer brings up the following considerations:

“1/ Consult the dMIQE2020 update guidelines and checklist for dPCR experiment reporting to inspect whether or not additional information could be provided.”

We thank the reviewer for this helpful suggestion, which guided us to expand the methodological description and to align more closely with the dMIQE2020 guidelines. Specifically, we added detailed information on nucleic acid extraction, quantification and storage, amplicon lengths and HXB2 genomic coordinates for Q4ddPCR primer/probe sets even when previously published, additional details on assay optimization using standard material (J-Lat 6.3 DNA), descriptions of analytical sensitivity, analytical specificity, technical repeatability and reproducibility and representative examples of positive and negative two-dimensional droplet plots.

“2/ Assay performance characteristics and metrics could be extended to determine LOB, LOD, LOQ, resolution between clusters. Are there dilution series performed on J-Lats? If not, would advise to include this. If sufficient NTCs were already run a LOB calculation could also be included. At the end the authors evaluate specificity and sensitivity but this could be confusing as these also can refer to the more technical aspects of an assay.”

We appreciate the reviewer's recommendation to expand the technical performance metrics. We added new Extended Data Figures and Tables summarizing the assessment of linearity and dynamic range, repeatability and reproducibility as well as analytical sensitivity and specificity.

We performed dilution series using J-Lat 6.3 DNA, but because HIV proviral templates in clinical samples frequently contain sequence mismatches that reduce signal amplitude and cluster separation, we complemented these experiments with full-process (from thawing PBMCs, isolating CD4⁺ T cells, extracting DNA to performing the assay itself) reproducibility measurements in four participants with lower reservoir sizes. These experiments confirmed internal consistency and close agreement between the Q4-based and IPDA-based Q4ddPCR configurations.

We also clarified throughout the manuscript when we refer to analytical sensitivity/specificity (e.g., LOB, LOD) versus the biological sensitivity/specificity describing the predictive capacity of Q4ddPCR to classify intact proviruses in clinical samples despite HIV sequence heterogeneity.

“3/ There is competition between the ENV-GAG and the GAG-POL, were any optimization approaches tried to further orthogonalize the position of the clusters?”

We thank the reviewer for this comment. We agree that maximizing cluster separation is critical for robust gating of multi-probe combinations, particularly in the context of the HIV sequence heterogeneity where in some samples primer/probe mismatches may result in reduced amplitude rather than complete extinction of signal.

To address this, we performed extensive optimization during assay development.

Specifically, we evaluated:

- Probe concentration titrations for all four targets in both Q4ddPCR variants
- Alternative fluorophore combinations to minimize spectral bleed-through and maximize cluster separation
- Probe modifications (locked nucleic acids or minor groove binders (MGB)) which improved cluster resolution only for the MGB-modified IPDA- ψ and *env*-probes (including the unlabeled hypermutant probe). In contrast, modifying the *gag* or *pol* probes did not substantially improve orthogonality in the *gag-pol* and *env-gag* clusters.
- Thermal cycling parameters, including annealing temperatures and number of cycles

As part of our design goal to maintain backwards compatibility with IPDA, we did not alter IPDA primers/probes nor their fluorophores.

We primarily optimized the assay on standardized material (J-Lat 6.3 DNA) but also on clinical samples, since cluster behavior can differ substantially in the presence of sequence heterogeneity. The final probe concentrations, fluorophore assignments and cycling conditions represent the configuration that maximized resolution across both standard material and clinical samples.

“4/ In terms of visualization, it would be helpful to color the true 4 color positive partitions in the 2D or 3D plots, so a visual inspection can be done of where these partitions lay. This can help in thresholding these out as the abovementioned competition could result in partitions that are close to the single positive cluster.”

We agree that visualizing the 4-color-positive droplets would be extremely valuable for assessing their position relative to negative clusters. We have consulted with Bio-Rad about this, as we were also interested in

generating such visualizations. Unfortunately, the current version of QX Manager does not allow 4-channel visualization of droplets within the same plot. To address this limitation, our gating strategy uses all six possible two-channel combinations, which allows us to infer the location of 4-target-positive droplets. In experiments with J-Lat 6.3 DNA, we observed that most 4-target-positive droplets clearly separate from the negative clusters. However, we also acknowledge that a subset can lie close to the negative population, particularly in the *pol*-channel (Fig. 1b, Extended Data Fig. 1c).

“5/ Add comment on potential subtype performance of the Q4ddPCR (subtype B oriented) or highlight flexible redesign opportunities of these type of assays to accommodate other subtypes.”

We thank the reviewer for raising this important point that has been raised by Reviewer 2 as well. Q4ddPCR, like IPDA, is currently optimized for subtype B, and its intactness calling relies on the same *env* probe used in IPDA. However, because Q4ddPCR incorporates built-in backup options (alternative probes and readouts), it is inherently more robust to subtype-related sequence variation than a strictly two-target assay. An additional advantage is the modular design of Q4ddPCR. Primer/probe sets can be readily exchanged and subtype-specific variants such as the published subtype B and C IPDA primer/probes can be directly integrated into the Q4ddPCR. This provides a practical path for adapting the method to non-B subtypes. However, validation on non-B samples will be required before drawing definitive conclusions about subtype performance.

“6/ On the Decision tree. How does it compare to other decision trees that are published? Would add section on this.”

We thank the reviewer for raising this point. We compared our decision tree to other published frameworks as part of our response to Reviewer 1 and 2.

“7/ Would the authors have data to support or hint towards the merit of adding more regions for intactness readouts and if there would be a limit in added value of performing for instance a 10 plex reaction. In other words, where is the limit?”

We thank the reviewer for this question. To explore the theoretical benefit of adding additional targets, we used our sequence-resolved dataset from 13 PWH for whom Q4PCR data were available. We constructed a mathematical model that estimated the expected sensitivity and specificity of hypothetical multi-probe combinations by using the experimentally observed performance of each single probe (see figure below). As anticipated, increasing the number of targets improved specificity for detecting intact proviruses (**a, d**). However, this gain exhibited saturation between four and five probes, while sensitivity (i.e., the probability that all probes would test positive in this case) declined progressively as more probes were added (**b, e**). The combination of sensitivity and specificity reaches an optimal point with 4 targets (**c**).

We evaluated this model under two scenarios: (1) assuming no primer/probe mismatches (**a-c**) and (2) incorporating the mismatches observed in our dataset (**d, e**). In the second, more realistic scenario, sensitivity dropped sharply with each additional probe. This finding further strengthens the need for backup strategies such as alternative probes/readouts rather than simply increasing the number of primary probes. But it has also to be noted that this model is simplified as it does not take into account the different specificity that specific probe combinations might have (e.g. the higher specificity of *env-Ψ*-positive combinations over *Ψ-gag*-positive combinations).

From a practical perspective, we also found technical barriers to implementing 5- or 6-plex ddPCR, including fluorophore bleed-through, particularly problematic when partial probe mismatches cause reduced (rather than absent) probe amplitudes. Finally, we did not include a 5'LTR probe because our Q4ddPCR readouts were already highly concordant with total HIV DNA quantified by a 5'LTR assay (Extended Data Fig. 4).

Taken together, both modeling and experimental constraints suggest diminishing benefits beyond four well-performing targets, with increasing risk of sensitivity loss, technical artefact and mismatch-driven dropout when moving toward very high-plex assays.

“8/ The authors mention 40K cells as minimal input. Could the authors elaborate on this threshold number or other QC for input? Also, if a 0 results is present, could they comment on the absence of signal not meaning absence of intact virus.”

We thank the reviewer for raising this important point. The threshold of 40,000 cell equivalents emerged from practical constraints associated with the ACTG A5321 cohort, in which sample material was very limited and with the aim of designing an assay that is suitable for other cohorts where sample size is limited e.g. pediatric cohorts or samples derived from mouse models. Because intact frequencies are normalized per million cells, lower DNA inputs inherently increase the risk of overestimation, particularly when 4-color droplets are rare. To evaluate the magnitude of this effect, we computationally simulated how identical droplet counts would translate into intact frequencies when assuming 20,000, 30,000, 40,000, 80,000, or 200,000 cell equivalents (the latter being our usual input).

As expected, estimates became increasingly inflated as input dropped below 40,000 cell equivalents, particularly in individuals with low reservoir sizes (10 - 30 intact/ 10^6 cells, see below, each dot represents one replicative well). Nevertheless, the deviation at 40,000 remained within an interpretable range for most samples. Based on these analyses, we recommend aiming for 150,000 - 200,000 cell equivalents whenever possible and applying additional caution when interpreting longitudinal trends if input varies substantially between time points. Monitoring cell equivalents via RPP30 thus serves as an important quality control criterion.

Regarding 0 values, we agree with the reviewer that the absence of detectable 4-color positive proviruses could theoretically mean that there are no intact proviruses present. This issue is intrinsic to all reservoir assays. We considered this carefully when comparing QVOA and Q4ddPCR. In all 16 PWH we had successful viral outgrowth. In this sample set Q4-based Q4ddPCR detected intact proviruses in 11 cases through 4-color positives and the remainder through valid 3-color positives. Similarly, IPDA-based Q4ddPCR detected intact proviruses using 4-colors in 15, 3-colors in 4 and 2-colors in 1 case. These findings support the robustness of our decision tree as Q4ddPCR correlates with viral outgrowth when it is applied and supports the hypothesis that if 4- or even 3-color positives are absent it does not necessarily mean that there are no intact proviruses.

Therefore, in cases where zero results are obtained, we recommend (1) using alternative primer/probe sets to mitigate primer/probe mismatch associated dropout, (2) increasing cell equivalents where feasible and (3) combining different reservoir measurements before concluding true absence of intact virus

“9/ The authors might consider to rename to Q4dPCR in stead of Q4ddPCR, referring to all digital PCR systems rather than only droplet-based systems.”

We thank the reviewer for this suggestion. At this stage, we prefer to retain the name Q4ddPCR as the assay was developed, optimized and validated specifically on droplet-based digital PCR platforms which is well established and known in the field. But we certainly recognize that the underlying conceptual framework could be adapted in the future to other digital PCR systems.

“10/ The evaluation of Q4ddPCR with Q4PCR generated viral sequence data could skew/bias the absolute % of sensitivity and specificity as the same primer sets are used. A cautionary note could be added while interpreting these values.”

We thank the reviewer for highlighting this important point. Indeed, because Q4PCR relies on long-distance PCR, it is biased toward recovering defective proviruses and tends to under-represent full-length genomes. This bias can carry over into the performance evaluation of Q4ddPCR, since the sensitivity and specificity estimates are based on sequences that were themselves obtained using Q4PCR. As a result, our calculated specificity is likely conservative: the reference dataset tends to be enriched for defective proviruses, increasing the probability of underestimating the accuracy of Q4ddPCR readouts. We have added a cautionary note in the discussion.